# DRAG: Data Reconstruction Attack using Guided Diffusion

Wa-Kin Lei [1]    Jun-Cheng Chen [2]    Shang-Tse Chen [1]

## Abstract

With the rise of large foundation models, split inference (SI) has emerged as a popular computational paradigm for deploying models across lightweight edge devices and cloud servers, addressing data privacy and computational cost concerns. However, most existing data reconstruction attacks have focused on smaller CNN classification models, leaving the privacy risks of foundation models in SI settings largely unexplored. To address this gap, we propose a novel data reconstruction attack based on guided diffusion, which leverages the rich prior knowledge embedded in a latent diffusion model (LDM) pre-trained on a large-scale dataset. Our method performs iterative reconstruction on the LDM's learned image prior, effectively generating high-fidelity images resembling the original data from their intermediate representations (IR). Extensive experiments demonstrate that our approach significantly outperforms state-of-the-art methods, both qualitatively and quantitatively, in reconstructing data from deep-layer IRs of the vision foundation model. The results highlight the urgent need for more robust privacy protection mechanisms for large models in SI scenarios. Code is available at: https://github.com/ntuaislab/DRAG

## 1. Introduction

The rapid development of deep learning has revolutionized various aspects of daily life—from AI assistants to autonomous vehicles. However, the substantial computational resources required by these emerging models often hinder their deployment on edge devices. Therefore, offloading intensive computation to cloud servers has become a popular alternative. Following this paradigm, split inference (SI) (Kang et al., 2017) has emerged as one of the most promising solutions, as it balances computational and privacy concerns. This approach enables efficient utilization of cloud resources, reduces the computational burden on local devices, and facilitates the integration of complex models into everyday technologies by partitioning neural network computations between edge devices and cloud servers, with data processed locally before being sent to the server.

Despite its advantages, recent studies (He et al., 2019; Dong et al., 2022; Li et al., 2023; Xu et al., 2024; Sa et al., 2024) have uncovered significant privacy risks associated with SI, particularly in the form of data reconstruction attacks (DRA). In DRA, adversaries attempt to reconstruct clients' input data by exploiting the exchanged intermediate representations (IR) between clients and servers, posing serious threats that break users' privacy.

While these risks are established, the growing adoption of more powerful models, such as Vision Transformers (ViT) (Dosovitskiy et al., 2021), raises concerns about the effectiveness of existing defenses. ViTs have demonstrated superior performance across various vision tasks and are widely used in modern applications. Nevertheless, the privacy implications of deploying these models in SI settings remain underexplored.

In this paper, we address this gap by investigating privacy leaks in vision transformers in the context of SI. We propose a novel attack based on guided diffusion that effectively utilizes the prior knowledge captured by large latent diffusion models (LDM) (Rombach et al., 2022) pre-trained on large-scale datasets (e.g., Stable Diffusion) to reconstruct input data from deep-layer IR. Leveraging this prior knowledge, we successfully invert IR back to the original input data across various natural image datasets, revealing a critical privacy vulnerability in the SI framework. Additionally, we evaluate our attack on models equipped with existing defenses (Singh et al., 2021; Vepakomma et al., 2020) and show that input data can still be successfully reconstructed from deep-layer IR despite the defenses. Our key contributions are summarized as follows:

- We propose DRAG, a novel attack that exploits the prior knowledge captured by LDMs to reconstruct input data from deep-layer IR.

- Our attack can reconstruct high-quality images from

---

[1] National Taiwan University [2] Research Center for Information Technology Innovation, Academia Sinica. Correspondence to: Shang-Tse Chen <stchen@csie.ntu.edu.tw>.

*Proceedings of the $42^{nd}$ International Conference on Machine Learning*, Vancouver, Canada. PMLR 267, 2025. Copyright 2025 by the author(s).

widely used vision foundation models, specifically CLIP (Radford et al., 2021) and DINOv2 (Oquab et al., 2024), demonstrating that the privacy threat exists even in widely used general-purpose vision encoders.

- We explore defense strategies tailored for vision transformers to mitigate the threat of privacy leakage.

## 2. Related Work

### 2.1. Split Inference

Split inference (SI) (Kang et al., 2017) is a method aimed at speeding up inference and/or reducing power consumption in endpoint devices while ensuring data privacy. It has been widely studied in various applications, including computer vision tasks such as image classification, detection, and segmentation, as well as natural language processing tasks such as language understanding (Matsubara et al., 2022). In recent years, SI has also attracted attention for its role in generative AI, including LLMs and text-to-image generation (Ohta & Nishio, 2023).

Unlike the traditional cloud-based inference approaches that require transmitting raw data to servers, SI preserves privacy by sending only transformed, non-trivially interpretable IR to the cloud. Specifically, the model $f$ is partitioned into two parts: the client model $f_c : \mathcal{X} \to \mathcal{H}$ that maps input data $\mathbf{x}$ from input space $\mathcal{X}$ to IR space $\mathcal{H}$, and the server model $f_s : \mathcal{H} \to \mathcal{Y}$ that maps IR to output space $\mathcal{Y}$. The client model is deployed on the edge device, while the server model operates in the cloud. During inference, the private data $\mathbf{x}^*$ is first processed at the edge by $f_c$, producing the "smashed" data $\mathbf{h}^* = f_c(\mathbf{x}^*)$. This IR is then transmitted to the cloud, where the server completes the computation by inferring $y^* = f_s(\mathbf{h}^*)$. This approach, which provides a certain degree of privacy preservation for users, also leverages the abundant computational resources of cloud servers to accelerate inference, making it a feasible solution for applications requiring both privacy and low-latency predictions.

### 2.2. Data Reconstruction Attack (DRA)

In the context of SI, an adversary may extract private information by reconstructing the original input $\mathbf{x}^*$ from $\mathbf{h}^*$, as illustrated in Figure 1. Following He et al. (2019), we categorize existing DRAs based on the adversary's knowledge of the client model as follows:

**White-box attacks** assume complete knowledge of the architecture and parameters of the client model. This assumption has become increasingly reasonable with the rise of large models leveraging frozen, publicly available vision encoders (Liu et al., 2023; Chen et al., 2023). The early work of He et al. (2019) framed reconstruction as *regularized Maximum Likelihood Estimation* (rMLE), which optimizes

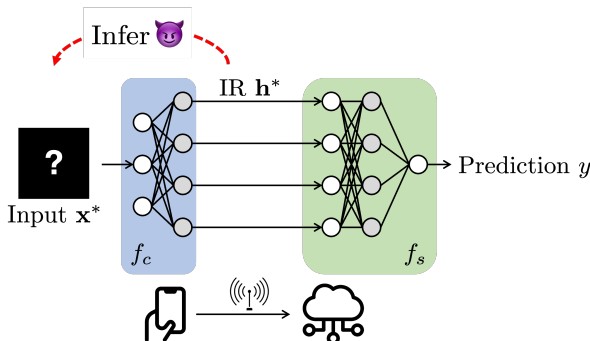

Figure 1: Privacy threats in split inference.

a candidate input to match the target IR, with Total Variation (Rudin et al., 1992) serving as an image prior. Singh et al. (2021) improved reconstruction quality by adding a deep image prior (Ulyanov et al., 2018) in their *Likelihood Maximization* (LM) method. More recently, Li et al. (2023) introduced *GAN-based Latent Space Search* (GLASS), which constrains reconstructions using StyleGAN2 (Karras et al., 2020b). This method yields high-fidelity images that successfully evade several defenses (He et al., 2019; Singh et al., 2021; Titcombe et al., 2021; Mireshghallah et al., 2020; Li et al., 2021; Osia et al., 2020).

**Black-box attacks** require only query access to $f_c$, typically utilizing inverse networks trained on input-output pairs (He et al., 2019). Recent work has enhanced this approach by incorporating diffusion models (Chen et al., 2024), which offer better reconstruction quality.

**Query-free attacks** operate without access to $f_c$. Instead, they use a collection of IRs to construct surrogate models $f_c'$ that approximate $f_c$, followed by applying reconstruction techniques to these surrogates. Beyond He et al. (2019), previous works (Pasquini et al., 2021; Erdoğan et al., 2022; Gao & Zhang, 2023; Xu et al., 2024) assume the adversary participates in the model training process to enhance the effectiveness of subsequent reconstruction attacks.

Existing evaluations primarily focused on CNN architectures like ResNet18 (He et al., 2016). However, ViTs process images fundamentally differently through patch tokenization and attention mechanisms, and their vulnerability to reconstruction attacks remains largely unexplored. Our analysis in Section 5.5 reveals that ViTs exhibit token order invariance, a feature absent in CNNs, which significantly affects attack effectiveness.

### 2.3. Diffusion Models

In recent years, diffusion models (Ho et al., 2020; Song et al., 2020) have demonstrated remarkable capabilities in generating realistic images by iteratively refining noise into coherent visuals. Conditional generation has been signif-

icantly advanced through methods such as classifier guidance (Dhariwal & Nichol, 2021) and classifier-free guidance (Ho & Salimans, 2021), which incorporate class information to guide the generation process. Specifically, classifier guidance (Dhariwal & Nichol, 2021) estimates the class probability $p_t(\mathbf{c}|\mathbf{x})$ while maintaining a fixed unconditional distribution $p_t(\mathbf{x})$, enabling conditional generation through the Bayes rule: $p_t(\mathbf{x}|\mathbf{c}) \propto p_t(\mathbf{x}) \, p_t(\mathbf{c}|\mathbf{x})$. Universal Guidance Diffusion (UGD) (Bansal et al., 2024) further expands the scope of conditional generation by integrating various neural networks to incorporate diverse conditions, such as object bounding boxes, segmentation masks, face identities, and stylistic attributes.

While classifier guidance seeks to adapt a fixed unconditional model $p_t(\mathbf{x})$ by developing an appropriate conditional distribution $p_t(\mathbf{c}|\mathbf{x})$, another line of research (Chung et al., 2023; Yang et al., 2024) assumes the availability of a predefined conditional distribution $p_0(\mathbf{c}|\mathbf{x})$. This assumption positions diffusion models as promising tools for addressing a range of conditional generation tasks.

Moreover, LDMs (Rombach et al., 2022) have further advanced diffusion models by enabling the generation of diverse, high-resolution, high-quality images within a latent space, thus improving computational efficiency. Subsequent work (Ramesh et al., 2022; Zhang et al., 2023) has extended control mechanisms within the latent diffusion framework, allowing more precise and hierarchical image manipulation.

# 3. Methodology

## 3.1. Threat Model

Following previous work (He et al., 2019; Singh et al., 2021; Dong et al., 2022; Li et al., 2023), we consider an honest-but-curious server in SI that seeks to reconstruct $\mathbf{x}^*$ from $\mathbf{h}^*$. Given the prevalence of frozen foundation models in downstream applications, we assume white-box access to the client model $f_c$, providing the adversary with complete architectural and parameter knowledge. Under this threat model, we formulate the problem using two complementary approaches: (1) optimization-based and (2) learning-based.

**Optimization Based.** The adversary aims to find input data $\mathbf{x}'$ whose corresponding IR closely matches $\mathbf{h}^*$ by solving the following optimization problem:

$$\mathbf{x}' = \arg\min_{\mathbf{x} \in \mathcal{X}} d_{\mathcal{H}}(f_c(\mathbf{x}), \mathbf{h}^*) + \lambda R_{\mathcal{I}}(\mathbf{x}), \qquad (1)$$

where $d_{\mathcal{H}}$ measures the distance between the IRs, $R_{\mathcal{I}}$ represents the regularization term ensuring perceptually realistic, and $\lambda \geq 0$ controls the weight of regularization. Ultimately, the adversary's goal is to achieve $\mathbf{x}' \approx \mathbf{x}^*$.

**Learning Based.** The adversary infers $\mathbf{x}^*$ by training an inverse network $f_c^{-1} \colon \mathcal{H} \to \mathcal{X}$, where the network is trained using paired data $(\mathbf{x}, f_c(\mathbf{x}))$ from a public dataset $D_{\text{public}}$:

$$f_c^{-1} = \arg\min_{f_c^{-1}} \mathbb{E}\left[||f_c^{-1}(f_c(\mathbf{x})) - \mathbf{x}||_2\right]. \qquad (2)$$

The following subsections detail our reconstruction approach, as illustrated in Figure 2.

## 3.2. Data Reconstruction Attack using Guided Diffusion

In this section, we propose leveraging guided diffusion for data reconstruction attacks. Pre-trained diffusion models serve as image priors $R_{\mathcal{I}}$, constraining reconstructions to the natural image manifold. Based on a noise predictor $\epsilon_\theta$ and its corresponding noise scheduler $\{(\alpha_t, \sigma_t)\}_{t=1}^T$, unconditional DDIM sampling (Song et al., 2020) transitions sample $\mathbf{x}$ from timestep $t$ to $t-1$ as follows:

$$\begin{aligned} \mathbf{x}_{t-1} = &\sqrt{\alpha_{t-1}} \left( \frac{\mathbf{x}_t - \sqrt{1 - \alpha_t}\,\epsilon_\theta(\mathbf{x}_t)}{\sqrt{\alpha_t}} \right) \\ &+ \sqrt{1 - \alpha_{t-1} - \sigma_t^2}\,\epsilon_\theta(\mathbf{x}_t) \\ &+ \sigma_t\,\epsilon_t, \end{aligned} \qquad (3)$$

where $\epsilon_t \sim \mathcal{N}(\mathbf{0}, \mathbf{I})$ denotes Gaussian noise sampled from a standard normal distribution. For conditional image generation, classifier guidance (Dhariwal & Nichol, 2021) modifies the predicted noise by adding the gradient of a classifier's log-probability to the predicted noise:

$$\epsilon_\theta(\mathbf{x}_t, \mathbf{c}) = \epsilon_\theta(\mathbf{x}_t) - w\sqrt{1 - \alpha_t}\nabla_{\mathbf{x}_t} \log p_\theta(\mathbf{c}|\mathbf{x}_t), \qquad (4)$$

where $\mathbf{c}$ denotes the condition, and $w$ controls the strength of guidance. Extending classifier guidance, UGD (Bansal et al., 2024) replaces the classification loss with a general, differentiable loss $L(\mathbf{x}_t, \mathbf{c})$. For data reconstruction, we define the objective function as:

$$L(\mathbf{x}_t, \mathbf{c}) = d_{\mathcal{H}}(f_c(\mathbf{x}_t), \mathbf{h}^*). \qquad (5)$$

Notably, directly feeding $\mathbf{x}_t$ to $f_c$ for guidance computation is unreliable, as $f_c$ is typically trained only on clean images. To address this, Chung et al. (2023) and Bansal et al. (2024) estimate the guidance using the unique posterior mean $\hat{\mathbf{x}}_0$, a single-step denoised sample:

$$L(\hat{\mathbf{x}}_0, \mathbf{c}) = d_{\mathcal{H}}(f_c(\hat{\mathbf{x}}_0), \mathbf{h}^*), \qquad (6)$$

where $\hat{\mathbf{x}}_0$ is computed using Tweedie's formula with a single forward pass through the noise predictor $\epsilon_\theta$:

$$\hat{\mathbf{x}}_0 = \frac{\mathbf{x}_t - \sqrt{1 - \alpha_t}\,\epsilon_\theta(\mathbf{x}_t)}{\sqrt{\alpha_t}}. \qquad (7)$$

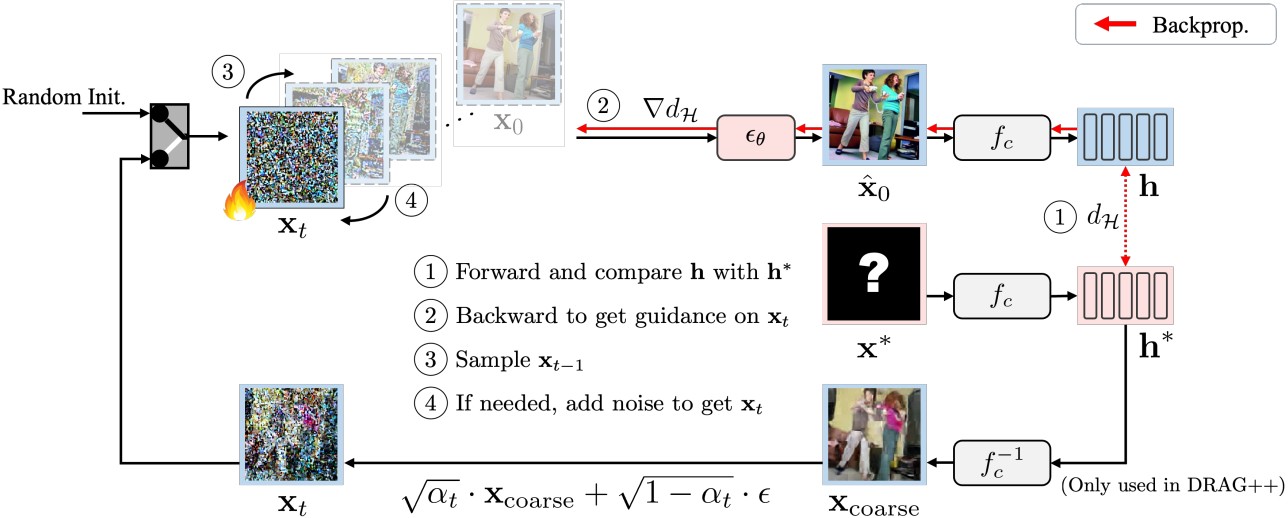

Figure 2: Illustration of DRAG (Data Reconstruction Attack using Guided Diffusion). The diffusion model serves as an image prior, constraining the solution space of $\mathbf{x}$ in optimization-based DRAs. Moreover, DRAG can be extended by incorporating the Inverse Network (He et al., 2019): we first obtain an initial estimate $\mathbf{x}_{\text{coarse}} = f_c^{-1}(\mathbf{h}^*)$, then refine it by diffusion-denoising process. We refer to this enhanced variant as DRAG++.

Diffusion with Spherical Gaussian constraint (DSG) (Yang et al., 2024) reduces the required denoising steps while enhancing generation quality. DSG aligns $\epsilon_t$ with the guidance $\mathbf{g}_t = \nabla_{\mathbf{x}_t} L(\hat{\mathbf{x}}_0, \mathbf{c})$ through:

$$\epsilon_t \leftarrow r \cdot \text{UNIT}((1-w)\,\sigma_t\,\epsilon_t + wr \cdot \text{UNIT}(\mathbf{g}_t)). \quad (8)$$

where $r = \sqrt{n}\sigma_t$ with $n$ being the dimension of $\mathbf{x}_t$, and operator $\text{UNIT}()$ normalizes vectors to unit norm. We adopt Equation (8) for our guided diffusion process.

To enhance reconstruction quality, we refine the guidance $\mathbf{g}_t$ using gradient clipping and historical guidance vector with optimizers such as Adam (Kingma & Ba, 2015). Since $f_c$ is typically non-convex, a single guidance step is insufficient for high-quality reconstruction. Therefore, we employ self-recurrence (Bansal et al., 2024), projecting $\mathbf{x}_{t-1}$ back to $\mathbf{x}_t$ via small-step DDPM diffusion (Ho et al., 2020) and iterating this denoising-diffusion cycle $k$ times:

$$\mathbf{x}_t = \sqrt{\alpha_t/\alpha_{t-1}} \cdot \mathbf{x}_{t-1} + \sqrt{1 - \alpha_t/\alpha_{t-1}} \cdot \epsilon. \quad (9)$$

We refer to this approach as DRAG—Data Reconstruction Attack using Guided Diffusion. Algorithm 1 outlines our proposed attack in detail. Note that the problem differs from previous works (Chung et al., 2023; Yang et al., 2024) in three key aspects, introducing additional challenges: (1) client models $f_c$ are typically assumed to be non-convex, (2) defensive mechanisms may be deployed, forcing attackers to operate in adversarial settings, (3) clients can embed randomness into $f_c$, further complicating the problem, as detailed in Section 5.5.

### 3.3. Extending DRAG with Inverse Networks

To enhance the performance and efficiency of DRAG, we integrate an auxiliary Inverse Network (He et al., 2019). This network accelerates DRAG by providing a coarse reconstruction from the target IR, serving as a more effective initialization. We train this network on an auxiliary dataset of publicly available data $D_{\text{public}}$ to project the IR back to image space and obtain $\mathbf{x}_{\text{coarse}}$. This coarse reconstruction is further projected onto an editable manifold by adding random noise at a small timestep $t = sT$:

$$\mathbf{x}_t = \sqrt{\alpha_t} \cdot \mathbf{x}_{\text{coarse}} + \sqrt{1 - \alpha_t} \cdot \epsilon, \text{ where } \epsilon \sim \mathcal{N}(\mathbf{0}, \mathbf{I}), \quad (10)$$

where the strength parameter $s$ satisfies $0 \leq s \leq 1$. We refer to this enhanced method as DRAG++.

### 3.4. Adapting to Latent Diffusion Models

As LDMs perform diffusion and denoising processing in the latent space $\mathcal{Z}$ instead of the pixel space $\mathcal{X}$, we adapt our approach when leveraging LDMs as the image prior by replacing the noisy sample $\mathbf{x}_t$ with the noisy latent $\mathbf{z}_t$. The mapping between $\mathcal{X}$ and $\mathcal{Z}$ is provided by the corresponding latent autoencoder $\mathcal{E}$ and $\mathcal{D}$. All other components of the method remain unchanged.

## 4. Experimental Setups

In this section, we first introduce the details of our experimental settings, including datasets, target models, compared methods, evaluation metrics, and attacker models.

**Algorithm 1** DRAG

// Noise $\epsilon$ is sampled from $\mathcal{N}(\mathbf{0}, \mathbf{I})$ for every usage
$\mathbf{s} \leftarrow \{ .\mathbf{m} = \mathbf{0}, .\mathbf{v} = \mathbf{0}, .i = 0 \}$
$\mathbf{z}_T \sim \mathcal{N}(\mathbf{0}, \mathbf{I})$
**for** $t = T$ **to** 1 **do**
  **for** $n = 1$ **to** $k$ **do**
    $\hat{\mathbf{x}}_0 \leftarrow \mathcal{D}(\text{TWEEDIESESTIMATION}(\mathbf{z}_t))$
    $\mathbf{g}_t \leftarrow \nabla_{\mathbf{z}_t}(d_{\mathcal{H}}(f_c(\hat{\mathbf{x}}_0), \mathbf{h}^*) + \lambda_{\ell_2} R_{\ell_2}(\hat{\mathbf{x}}_0))$
    $\overline{\mathbf{g}}_t \leftarrow \text{CLIPNORM}(\mathbf{g}_t, \mathbf{c}_{\max})$
    $\overline{\mathbf{g}}_t, \mathbf{s} \leftarrow \text{STATEUPDATE}(\overline{\mathbf{g}}_t, \mathbf{s})$
    $\mathbf{z}_{t-1} \leftarrow \text{GUIDEDSAMPLING}(\mathbf{z}_t, \overline{\mathbf{g}}_t)$
    $\mathbf{z}_t \leftarrow \sqrt{\alpha_t/\alpha_{t-1}} \cdot \mathbf{z}_{t-1} + \sqrt{1 - \alpha_t/\alpha_{t-1}} \cdot \epsilon$
  **end for**
**end for**
**return** $\mathcal{D}(\mathbf{z}_0)$

// Refine $\overline{\mathbf{g}}_t$ via momentum such as Adam
**function** STATEUPDATE($\overline{\mathbf{g}}_t, \mathbf{s}$)
  $\mathbf{s}.\mathbf{m} \leftarrow \beta_1 \cdot \mathbf{s}.\mathbf{m} + (1 - \beta_1) \cdot \overline{\mathbf{g}}_t$
  $\mathbf{s}.\mathbf{v} \leftarrow \beta_2 \cdot \mathbf{s}.\mathbf{v} + (1 - \beta_2) \cdot \overline{\mathbf{g}}_t^2$
  $\mathbf{s}.i \leftarrow \mathbf{s}.i + 1$
  $\hat{\mathbf{m}}, \hat{\mathbf{v}} \leftarrow \mathbf{s}.\mathbf{m}/\left(1 - \beta_1^i\right), \mathbf{s}.\mathbf{v}/\left(1 - \beta_2^i\right)$
  $\overline{\mathbf{g}}_t \leftarrow \hat{\mathbf{m}}/\left(\sqrt{\hat{\mathbf{v}}} + 10^{-8}\right)$
  **return** $\overline{\mathbf{g}}_t, \mathbf{s}$
**end function**

**function** GUIDEDSAMPLING($\mathbf{z}_t, \mathbf{g}_t$)
  $\epsilon_t \leftarrow r \cdot \text{UNIT}((1 - w)\, \sigma_t\, \epsilon + wr \cdot \text{UNIT}(\mathbf{g}_t))$
  **return** DDIM($\mathbf{z}_t, \epsilon_\theta(\mathbf{z}_t), \epsilon_t$)
**end function**

## 4.1. Datasets

To evaluate our proposed methods, we sample 10 images from the official validation splits of each dataset: (1) MSCOCO (Lin et al., 2014), (2) FFHQ (Karras et al., 2019), and (3) ImageNet-1K (Deng et al., 2009), constructing a collection of diverse natural images. All images are center-cropped and resized to 224×224. We use ImageNet-1K image classification as the primary task to quantitatively assess model utility. To simulate realistic conditions where the client and adversary have non-overlapping datasets, we randomly split the official training split of ImageNet-1K into two distinct, equal-sized and non-overlapping subsets: a private portion $D_{\text{private}}$ and a public portion $D_{\text{public}}$. The target model $f$ is fine-tuned exclusively on $D_{\text{private}}$, while the inverse network $f_c^{-1}$, as proposed in Section 3.3, is trained solely on $D_{\text{public}}$.

## 4.2. Target Models

We aim to reconstruct data from the widely used CLIP-ViT-B/16, CLIP-RN50 (Radford et al., 2021), and DINOv2-Base

(Oquab et al., 2024) vision encoder, known for its strong adaptability and zero-shot capabilities across vision tasks (Rao et al., 2022; Mokady et al., 2021). The reconstruction is performed after layers $l = \{0, 3, 6, 9, 12\}$ for CLIP-ViT-B/16 and DINOv2-Base, while for CLIP-RN50, the attack is conducted after blocks $l = \{1, 2, 3, 4, 5\}$. We evaluated the attack in three configurations: (1) the model is frozen at the pre-trained checkpoint, (2) protected by DISCO (Singh et al., 2021), and (3) protected by NoPeek (Vepakomma et al., 2020). The details of these two defenses can be found in Appendix D. These two defenses, highlighted in Li et al. (2023), have demonstrated superior privacy-preserving performance compared to other defenses.

## 4.3. Baseline and Metrics

We compare our method with previous optimization-based DRAs: rMLE (He et al., 2019), LM (Singh et al., 2021) and GLASS (Li et al., 2023). Implementation details are provided in Appendix C. To quantify privacy leakage across these attacks, we evaluated reconstruction performance using three complementary metrics: MS-SSIM (Wang et al., 2003), LPIPS (Zhang et al., 2018), and image similarity measured by DINO ViT-S/16 (Caron et al., 2021). These metrics capture both low-level fidelity and high-level semantic similarity, better reflecting privacy risks by aligning with human perceptual judgment compared to pixel-wise measures such as MSE or PSNR (Horé & Ziou, 2010).

## 4.4. Attacker Models

We use Stable Diffusion v1.5 (SDv1.5) as our image prior. GLASS (Li et al., 2023) uses domain-specific priors: StyleGAN2-ADA (FFHQ) (Karras et al., 2020a) for facial images and StyleGAN-XL (ImageNet-1K) (Sauer et al., 2022) for others, assuming knowledge of the target distribution. This inherently advantages GLASS by matching priors to true data distribution, while our diffusion-based approach uses a single, domain-agnostic prior.

For the architecture of the inverse network, we adopt the decoder architecture from He et al. (2022) to reconstruct images from the tokenized representations produced by ViTs. During training, we randomly replace $r_{\text{mask}} = 25\%$ of patch tokens with [MASK] token to improve its generalization.

## 4.5. Distance Metrics and Regularization

We use token-wise cosine distance as the distance metric $d_{\mathcal{H}}$ for the data reconstruction process in ViT-family models:

$$d_{\mathcal{H}}(\mathbf{h}_1, \mathbf{h}_2) = 1 - \frac{1}{N} \sum_{i=1}^{N} \frac{\langle \mathbf{h}_1[i, :], \mathbf{h}_2[i, :] \rangle}{||\mathbf{h}_1[i, :]|| \cdot ||\mathbf{h}_2[i, :]||}, \quad (11)$$

where $N$ denotes the number of tokens. For CLIP-RN50, the MSE loss is adopted. To prevent the latent $\mathbf{z}_t$ from

Table 1: Reconstruction performance of optimization-based attacks across target models and split points without defenses. **Bold** indicates the best scores, while underlined indicate the second-best.

| Method | Split Point | CLIP-ViT-B/16 | | | DINOv2-Base | | | Split Point | CLIP-RN50 | | |
|---|---|---|---|---|---|---|---|---|---|---|---|
| | | MS-SSIM ↑ | LPIPS ↓ | DINO ↑ | MS-SSIM ↑ | LPIPS ↓ | DINO ↑ | | MS-SSIM ↑ | LPIPS ↓ | DINO ↑ |
| rMLE | | 0.8888 | 0.0709 | 0.9712 | 0.9162 | 0.0504 | 0.9630 | | 0.6832 | 0.1543 | 0.9111 |
| LM | Layer 0 | **0.9638** | **0.0237** | **0.9903** | **0.9698** | **0.0227** | **0.9850** | Block 1 | **0.9769** | **0.0150** | **0.9919** |
| GLASS | | 0.8700 | 0.1466 | 0.8289 | 0.9147 | 0.1076 | 0.8369 | | 0.9052 | 0.0785 | 0.8485 |
| DRAG | | 0.9588 | 0.0489 | 0.9259 | 0.9567 | 0.0440 | 0.9284 | | 0.9316 | 0.0476 | 0.9454 |
| rMLE | | 0.8612 | 0.0914 | 0.9706 | 0.8371 | 0.1641 | 0.9302 | | 0.5741 | 0.2618 | 0.9053 |
| LM | Layer 3 | **0.9742** | **0.0206** | **0.9923** | **0.9682** | **0.0250** | **0.9890** | Block 2 | **0.9313** | **0.0382** | **0.9868** |
| GLASS | | 0.9091 | 0.0556 | 0.9623 | 0.9340 | 0.0488 | 0.9580 | | 0.7829 | 0.2185 | 0.7563 |
| DRAG | | 0.9500 | 0.0372 | 0.9715 | 0.9570 | 0.0323 | 0.9728 | | 0.9151 | 0.0604 | 0.9539 |
| rMLE | | 0.6888 | 0.2608 | 0.8875 | 0.6566 | 0.2562 | 0.9111 | | 0.6745 | 0.2233 | 0.8986 |
| LM | Layer 6 | 0.8604 | 0.0784 | 0.9734 | 0.7334 | 0.1676 | **0.9768** | Block 3 | 0.9028 | 0.0596 | **0.9769** |
| GLASS | | 0.7113 | 0.1352 | 0.9326 | 0.7444 | 0.1495 | 0.9333 | | 0.6785 | 0.2405 | 0.7877 |
| DRAG | | **0.9028** | **0.0465** | **0.9784** | **0.9196** | **0.0455** | 0.9782 | | **0.9118** | **0.0528** | 0.9662 |
| rMLE | | 0.4957 | 0.5131 | 0.7159 | 0.5855 | 0.4374 | 0.7663 | | 0.4888 | 0.4198 | 0.7776 |
| LM | Layer 9 | 0.6681 | 0.2138 | 0.9063 | 0.5281 | 0.3839 | 0.9555 | Block 4 | 0.5855 | 0.2576 | 0.9012 |
| GLASS | | 0.3852 | 0.4310 | 0.6740 | 0.5404 | 0.3230 | 0.8467 | | 0.4872 | 0.3568 | 0.7315 |
| DRAG | | **0.7974** | **0.0967** | **0.9652** | **0.8483** | **0.0820** | **0.9719** | | **0.7896** | **0.0898** | **0.9622** |
| rMLE | | 0.3884 | 0.5900 | 0.6524 | 0.4375 | 0.5680 | 0.6079 | | 0.3980 | 0.5006 | 0.6739 |
| LM | Layer 12 | 0.2560 | 0.6024 | 0.4248 | 0.3640 | 0.6190 | 0.8878 | Block 5 | 0.4432 | 0.3409 | 0.7614 |
| GLASS | | 0.2396 | 0.5790 | 0.4553 | 0.4456 | 0.4076 | 0.7297 | | 0.2917 | 0.4223 | 0.6811 |
| DRAG | | **0.6735** | **0.1857** | **0.9331** | **0.7581** | **0.1443** | **0.9463** | | **0.5206** | **0.2231** | **0.9001** |

deviating too far from the domain of the noise predictor $\epsilon_\theta$, we introduce $\ell_2$ regularization on $\hat{\mathbf{x}}_0$ to the objective (Equation (6)) to ensure it remains within the range $[-1, 1]$ during the reconstruction process:

$$R_{\ell_2}(\hat{\mathbf{x}}_0) = \frac{\lambda_{\ell_2}}{\text{CHW}} \cdot \hat{\mathbf{x}}_0^2. \qquad (12)$$

## 5. Experimental Results

In this section, we first compare the reconstruction performance of DRAG with prior methods on frozen, pre-trained foundation models. Next, we evaluate DRAG++, which integrates an auxiliary inverse network (IN) to improve reconstruction performance. Subsequently, we evaluate the generalization ability of DRAG on out-of-distribution data. We then examine its robustness against two defenses, DISCO and NoPeek. Finally, we analyze the reconstruction performance when applying the token shuffling defense, a mechanism intrinsic to ViTs. Complete experimental results can be found in Appendix A, and further experiments on the effects of key hyperparameters appear in Appendix B.

### 5.1. Reconstruction from Frozen Foundation Models

Table 1 presents quantitative reconstruction results, while Figure 3 visualizes results on CLIP-ViT-B/16, highlighting qualitative differences. Our approach consistently outperforms prior methods in reconstructing data from deeper layers. Although rMLE and LM achieve competitive performance at shallow split points, their reconstruction quality

degrades beyond layer 9 and layer 12, respectively. In contrast, DRAG outperforms the others at deeper split points across the evaluated metrics.

Compared to GLASS, which also utilizes a data-driven image prior and achieves strong performance on the FFHQ dataset, our method attains comparable performance on FFHQ and generalizes robustly to MSCOCO and ImageNet. Despite employing a GAN trained on ImageNet, GLASS reconstructs images from ImageNet with evident artifacts at deep split points. In contrast, DRAG maintains high-fidelity outputs free from such distortions, as shown in Figure 3.

### 5.2. Enhancing DRAG with Inverse Networks

We evaluate DRAG++ on CLIP-ViT-B/16 by comparing its performance to the inverse network (IN) and the original DRAG method. As shown in Table 2, DRAG++ achieves higher reconstruction performance in terms of LPIPS and DINO at split points beyond layer 3. While IN attains higher MS-SSIM, DRAG++ demonstrates notable improvements in LPIPS and DINO, with only a slight sacrifice in MS-SSIM compared to IN. This result suggests that the combined approach enhances perceptual quality in reconstruction.

### 5.3. Distribution Shift

We evaluate the generalization capability of DRAG and DRAG++ through two experiments in out-of-distribution settings. In the first experiment, we reconstruct aerial images from the UCMerced LandUse dataset (Yang &

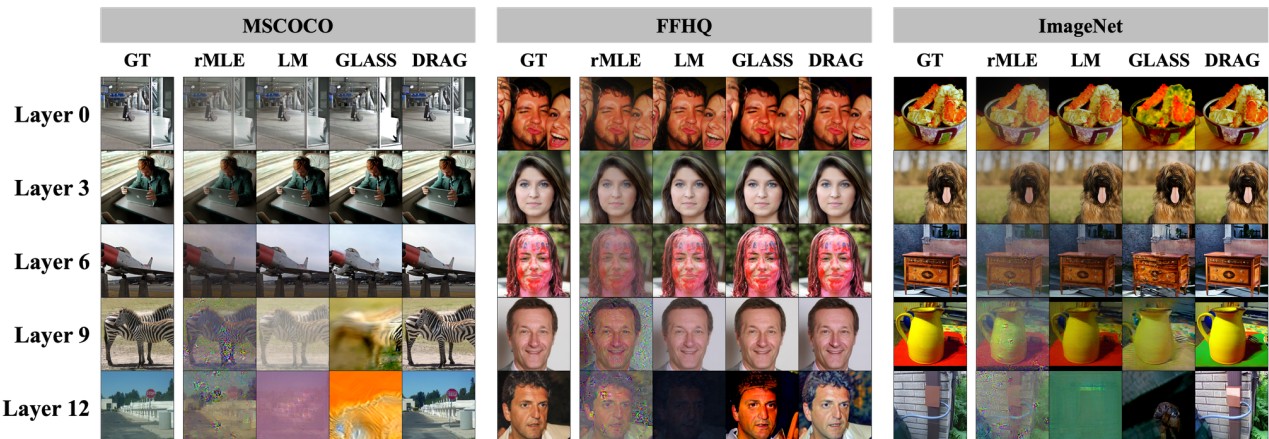

Figure 3: Reconstruction results for CLIP-ViT-B/16 across split points without defenses.

Table 2: Reconstruction performance comparison between inverse network alone and DRAG++ on CLIP-ViT-B/16.

| Split Point | Method | MS-SSIM ↑ | LPIPS ↓ | DINO ↑ |
|---|---|---|---|---|
| Layer 0 | IN | **0.9907** | **0.0112** | **0.9937** |
| | DRAG | 0.9588 | 0.0489 | _0.9259_ |
| | DRAG++ | _0.9608_ | _0.0485_ | 0.9234 |
| Layer 3 | IN | **0.9763** | _0.0351_ | 0.9458 |
| | DRAG | 0.9500 | 0.0372 | _0.9715_ |
| | DRAG++ | _0.9504_ | **0.0349** | **0.9719** |
| Layer 6 | IN | **0.9120** | 0.1799 | 0.7869 |
| | DRAG | 0.9028 | _0.0465_ | _0.9784_ |
| | DRAG++ | _0.9093_ | **0.0457** | **0.9785** |
| Layer 9 | IN | _0.8188_ | 0.2993 | 0.7130 |
| | DRAG | 0.7974 | _0.0967_ | _0.9652_ |
| | DRAG++ | **0.8224** | **0.0875** | **0.9700** |
| Layer 12 | IN | **0.7443** | 0.3618 | 0.6660 |
| | DRAG | 0.6735 | _0.1857_ | _0.9331_ |
| | DRAG++ | _0.7257_ | **0.1685** | **0.9492** |

Table 3: Reconstruction performance using diffusion model trained on out-of-distribution data for CLIP-ViT-B/16.

| Split Point | Method | MS-SSIM ↑ | LPIPS ↓ | DINO ↑ |
|---|---|---|---|---|
| Layer 0 | rMLE | 0.8888 | 0.0709 | _0.9712_ |
| | LM | _0.9638_ | **0.0237** | **0.9903** |
| | GLASS | 0.8700 | 0.1466 | 0.8289 |
| | DRAG[†] | **0.9692** | _0.0275_ | 0.9591 |
| Layer 3 | rMLE | 0.8612 | 0.0914 | 0.9706 |
| | LM | **0.9742** | **0.0206** | **0.9923** |
| | GLASS | 0.9091 | 0.0556 | 0.9623 |
| | DRAG[†] | _0.9488_ | _0.0427_ | _0.9722_ |
| Layer 6 | rMLE | 0.6888 | 0.2608 | 0.8875 |
| | LM | **0.8604** | **0.0784** | **0.9734** |
| | GLASS | 0.7113 | 0.1352 | 0.9326 |
| | DRAG[†] | _0.8244_ | _0.1574_ | _0.9438_ |
| Layer 9 | rMLE | 0.4957 | 0.5131 | 0.7159 |
| | LM | **0.6681** | **0.2138** | **0.9063** |
| | GLASS | 0.3852 | 0.4310 | 0.6740 |
| | DRAG[†] | _0.5379_ | _0.3937_ | _0.8154_ |
| Layer 12 | rMLE | _0.3884_ | 0.5900 | _0.6524_ |
| | LM | 0.2560 | 0.6024 | 0.4248 |
| | GLASS | 0.2396 | _0.5790_ | 0.4553 |
| | DRAG[†] | **0.3958** | **0.4938** | **0.7245** |

[†] Based on diffusion model trained on LSUN bedroom dataset.

Newsam, 2010), using $f_c^{-1}$ trained on ImageNet-1K. As shown in Figure 4, DRAG and DRAG++ successfully reconstruct these images, demonstrating their robustness across different domains. In the second experiment, we replace the diffusion model with a domain-specific model trained on the LSUN-bedroom dataset (Yu et al., 2015). We compare this configuration while keeping the other attack settings unchanged. The results shown in Table 3 indicate that DRAG achieves the best performance at layer 12 and ranks as the runner-up at layers 3, 6, and 9, further demonstrating its robustness even in an out-of-distribution setting.

### 5.4. Reconstruction from Privacy-Guarded Models

We evaluate attacks on models protected by DISCO and NoPeek defenses with varying hyperparameter settings that determine privacy protection strength, as detailed in Table 4.

Since Config-III and Config-VI represent the most challenging settings for adversaries, we highlight the results under these configurations in Table 5 and Figure 5. Our attack proves effective against weaker defense configurations, with comprehensive results across all configurations presented in Appendix A.2 and Figure 9.

For DISCO, we assume the adversary lacks knowledge of the pruning model or the mask applied to $\mathbf{h}^*$, as pruning is a dynamic, auxiliary component that can be decomposed from $f_c$. The pruned channels mislead the reconstruction

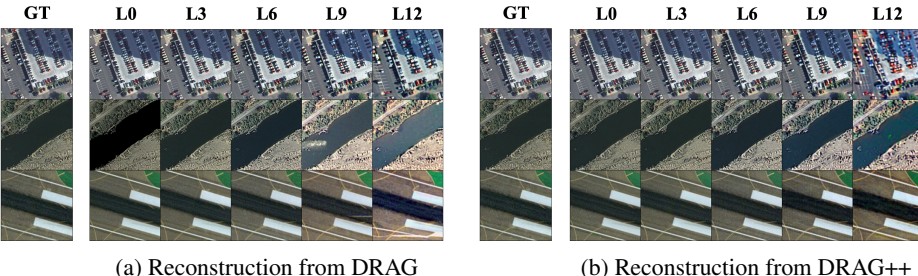

| GT | L0 | L3 | L6 | L9 | L12 | GT | L0 | L3 | L6 | L9 | L12 |

(a) Reconstruction from DRAG  (b) Reconstruction from DRAG++

Figure 4: Reconstruction results on out-of-distribution aerial images from the UCMerced LandUse dataset.

Table 4: Defense parameters for DISCO and NoPeek.

| Defense | Config | Defense Parameters |
|---------|--------|--------------------|
| DISCO | I | $\rho_D = 0.95, r_p = 0.1$ |
|  | II | $\rho_D = 0.75, r_p = 0.2$ |
|  | III | $\rho_D = 0.95, r_p = 0.5$ |
| NoPeek | IV | $\rho_N = 1.0$ |
|  | V | $\rho_N = 3.0$ |
|  | VI | $\rho_N = 5.0$ |

Table 5: Performance under DISCO (Config-III) and NoPeek (Config-VI) for CLIP-ViT-B/16 at layer 12.

| Method | MS-SSIM ↑ | LPIPS ↓ | DINO ↑ |
|--------|-----------|---------|--------|
| Config-III | | | |
| rMLE | **0.1686** | 0.8128 | 0.1129 |
| LM | 0.1638 | 0.7079 | 0.1654 |
| GLASS | 0.1479 | **0.6225** | **0.3878** |
| DRAG | 0.0788 | 0.7449 | 0.3201 |
| Config-III - w/ adaptive attacks | | | |
| rMLE | 0.2101 | 0.7822 | 0.2072 |
| LM | 0.1696 | 0.6953 | 0.1818 |
| GLASS | 0.2402 | 0.5401 | 0.4862 |
| DRAG | **0.4557** | **0.3778** | **0.7557** |
| Config-VI | | | |
| rMLE | 0.2270 | 0.7048 | 0.3747 |
| LM | 0.1783 | 0.7917 | 0.2099 |
| GLASS | 0.1950 | 0.7248 | 0.3141 |
| DRAG | **0.4469** | **0.3836** | **0.8096** |

process, leading to failures in the most challenging setting, Config-III. However, since pruned channels generally have smaller absolute values than unpruned counterparts, this discrepancy can be exploited by adaptive attacks. Specifically, the adversary can filter out channels with low mean absolute values when calculating $d_{\mathcal{H}}$, thereby mitigating the misleading influence of pruned channels. Our experimental results also show that DRAG performs the best after the adaptive attack filters out the pruned channels.

For NoPeek, we observe that $d_{\mathcal{H}}$ is significantly lower than in unprotected models during the optimization process, consistent with findings in Li et al. (2023). Despite this, DRAG still reconstructs the target images with higher fidelity compared to the previous works.

### 5.5. Token Shuffling (and Dropping) Defense

ViTs naturally exhibit an adaptive computation capability, enabling reduced inference time by discarding redundant tokens in the intermediate layers. Previous work (Yin et al., 2022) investigates strategies for dropping tokens in intermediate layers. From a privacy protection perspective, shuffling (and dropping) patch tokens hinders data reconstruction, as the distance metrics $d_{\mathcal{H}}$ is sensitive to the token order. For tasks where token order is irrelevant (e.g., classification), shuffling patch tokens offers a straightforward defense against DRA. Additionally, this method is easy to implement, as it only requires memory copying.

We evaluate the privacy risk against the token-shuffling (and dropping) defense. To simulate a token-dropping scenario,

we propose the following protocol: the client shuffles the patch tokens and randomly drops $r_{\text{drop}} N$ patch tokens before sending them to the server, where $r_{\text{drop}}$ is the proportion of the dropped tokens. Specifically, shuffling is applied after the positional embedding layer, while the [CLS] token remains fixed in position.

As noted in Darcet et al. (2024), tokens retain information about their original positions, which can be inferred by a classifier. Based on this observation, we train a 2-layer MLP classifier to predict the probability that a token $\mathbf{h}[i, :]$ was originally at position $\arg\max p_\theta(\mathbf{h}[i, :])$. Once trained, the classifier enables us to reorder the patch tokens by solving an assignment problem, maximizing the joint probability using the Hungarian algorithm (Kuhn, 1955).

We present the reconstruction results under three configurations: (1) patch tokens are randomly permuted, and the adversary is unaware of the permutation; (2) the adversary uses a token position classifier to reorder the tokens; and

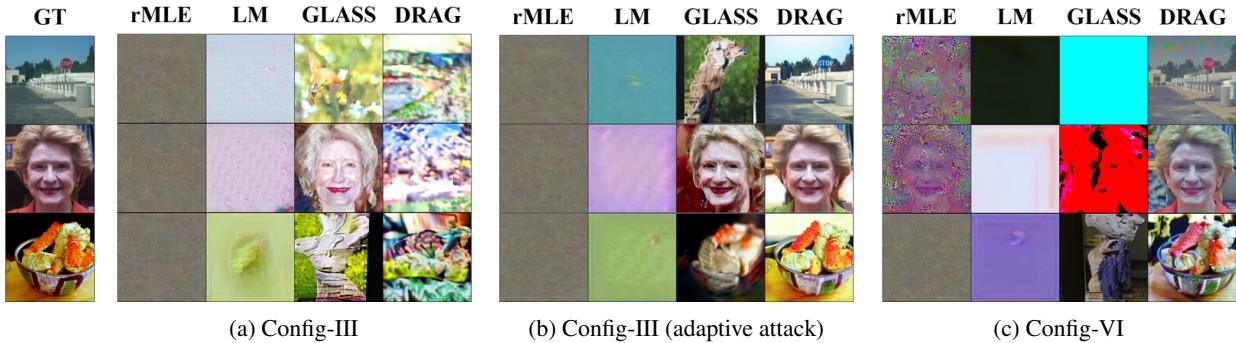

(a) Config-III

(b) Config-III (adaptive attack)

(c) Config-VI

Figure 5: Reconstruction results for CLIP-ViT-B/16 at layer 12 with DISCO (Config-III) and NoPeek (Config-VI).

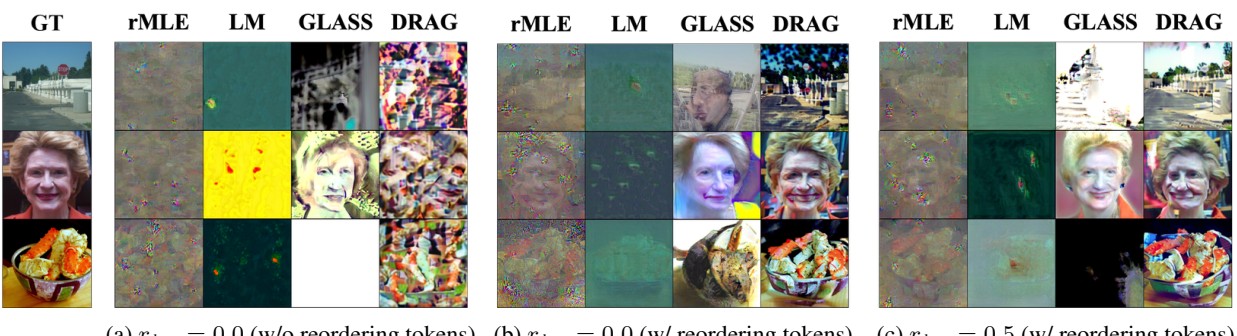

(a) $r_{\text{drop}} = 0.0$ (w/o reordering tokens)   (b) $r_{\text{drop}} = 0.0$ (w/ reordering tokens)   (c) $r_{\text{drop}} = 0.5$ (w/ reordering tokens)

Figure 6: Reconstruction results for CLIP-ViT-B/16 at layer 12 with token shuffling defense.

(3) the client drops 50% of the patch tokens before sending them to the server, leaving the adversary to infer their correct placement. Experiments are conducted on CLIP-ViT-B/16 splitting at layer 12. The token position classification model achieves 21.40% top-1 accuracy in predicting token positions, with an average $\ell_1$ distance of 2.34 from the correct position on ImageNet-1K. As shown in Figure 6, both rMLE and LM fail to reconstruct the target images, whereas the normal configuration succeeds. For GLASS and DRAG, reconstruction performance is degraded in the shuffled scenarios, but some reconstructed images still retain key features of the original inputs.

## 6. Conclusion

This work reveals significant privacy risks in SI with large vision foundation models like CLIP-ViT and DINOv2, extending beyond previous attacks on CNN models like ResNet18. We propose a novel data reconstruction attack leveraging LDMs pre-trained on large-scale datasets. Our method generates high-fidelity images from IR and outperforms state-of-the-art approaches in reconstructing data from deep-layer IR. These findings highlight the need for stronger defenses to protect privacy when deploying transformer-based models in SI settings.

## Acknowledgements

This research is supported by National Science and Technology Council, Taiwan under grant numbers 113-2222-E-002-004-MY3, 113-2634-F-002-007, and 113-2634-F-002-001-MBK; by the Featured Area Research Center Program within the Higher Education Sprout Project of the Ministry of Education (grant 113L900903); and by Academia Sinica (grant AS-CDA-110-M09). We also thank the National Center for High-performance Computing of National Applied Research Laboratories in Taiwan for providing computational and storage resources.

## Impact Statement

This work investigates the privacy risks of large vision foundation models in the split inference framework. As these models are widely adopted in downstream tasks, privacy concerns become crucial. Our proposed DRAG method shows that, unlike smaller CNN models studied previously, large vision models are also vulnerable to embedding reconstruction attacks, posing risks to applications like domestic robots and cyber-physical systems. These findings provide key insights for ML developers and users, urging a re-evaluation of privacy risks and the development of more robust, privacy-preserving architectures.

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

# A. Report

## A.1. Reconstruction from Frozen Foundation Models

Besides Table 1, we provide a figure in Figure 7 that visualizes the performance of each attack. Additionally, we present the attack success rate (ASR), which is defined as the proportion of images for which the reconstruction metrics exceed a specified threshold, as shown in Figure 7.

Figure 8 illustrates the performance of attacks on the same target images at different split points for CLIP-ViT-B/16. An image from each dataset was chosen for evaluation.

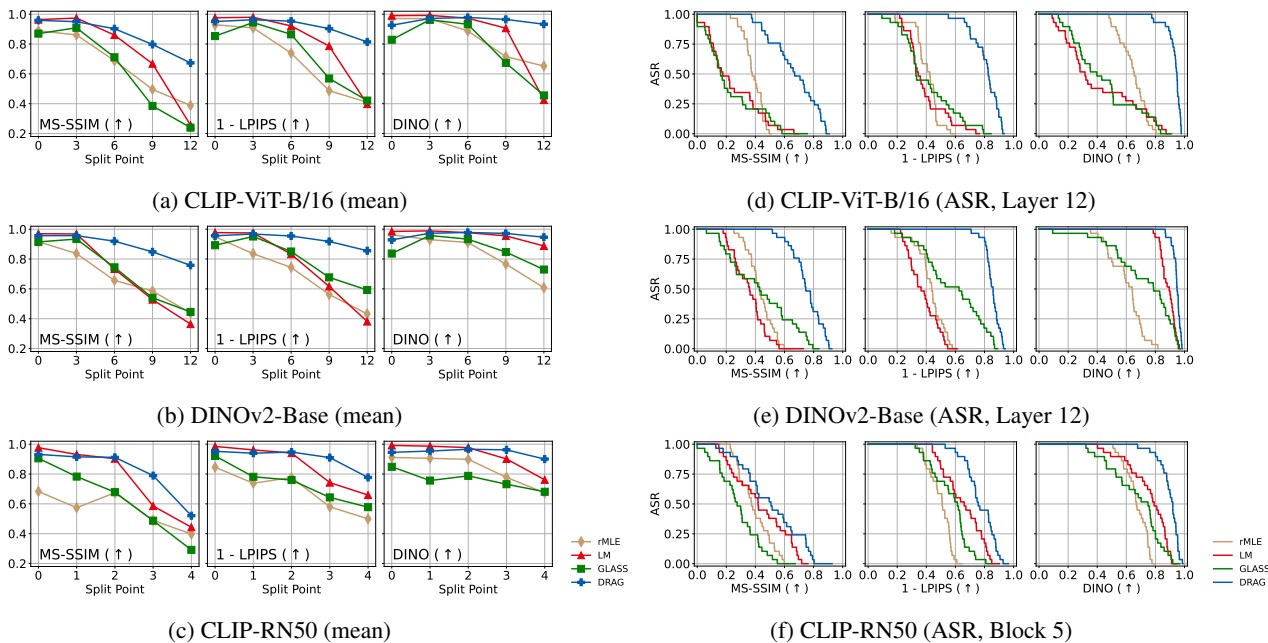

Figure 7: Reconstruction quality metric and attack success rate (ASR) across target models without defenses.

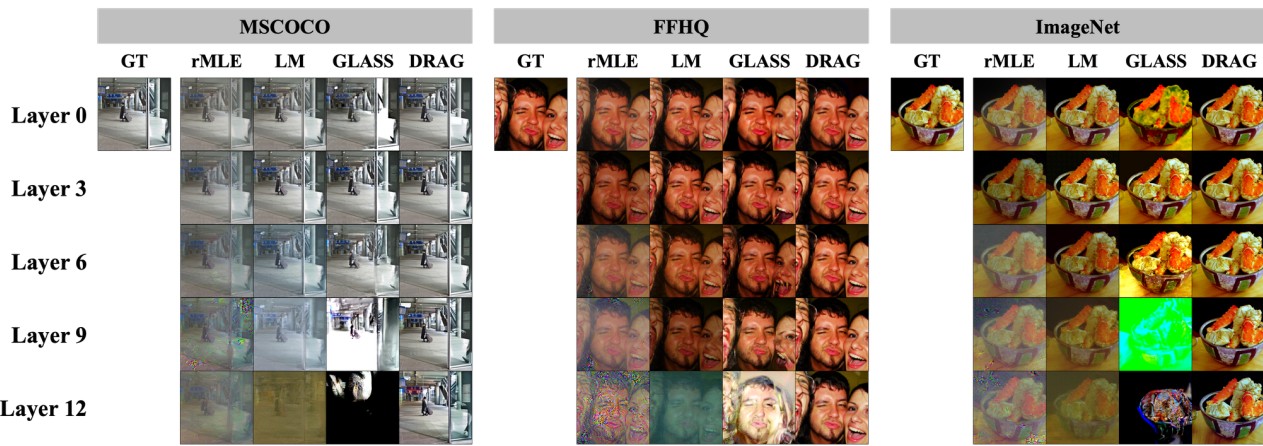

Figure 8: Reconstruction results for CLIP-ViT-B/16 across split points without defenses. The same images are used as evaluation targets to compare the performance of previous attacks.

## A.2. Reconstruction from Privacy Guarded Models

Table 6 provides the model utility, which is measured by classification accuracy on ImageNet-1K. The complete quantitative results for Section 5.4, conducted on CLIP-ViT-B/16 and DINOv2-Base, as presented in Table 7 and Figure 9.

Table 6: Model utility under privacy defenses, measured by ImageNet-1K classification accuracy.

| Defense | Config | Parameters | CLIP-ViT-B/16 | DINOv2-Base |
|---|---|---|---|---|
| w/o defense | | | 79.87% | 83.81% |
| DISCO | I | $\rho_D = 0.95, r_p = 0.1$ | 79.20% | 83.51% |
| | II | $\rho_D = 0.75, r_p = 0.2$ | 79.02% | 83.46% |
| | III | $\rho_D = 0.95, r_p = 0.5$ | 78.04% | 82.85% |
| NoPeek | IV | $\rho_N = 1.0$ | 79.28% | 83.43% |
| | V | $\rho_N = 3.0$ | 78.67% | 83.39% |
| | VI | $\rho_N = 5.0$ | 77.88% | 83.22% |

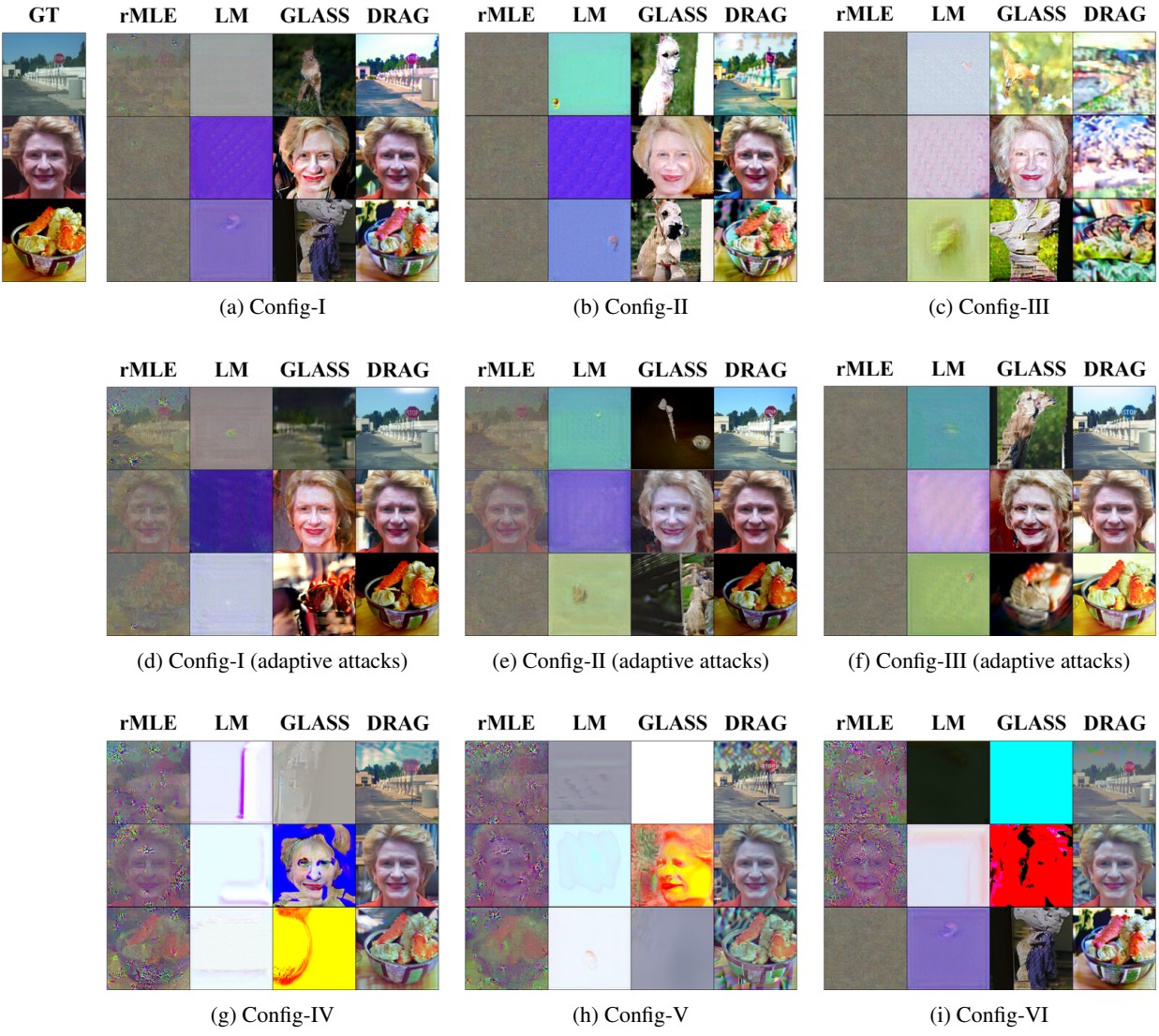

(a) Config-I  (b) Config-II  (c) Config-III

(d) Config-I (adaptive attacks)  (e) Config-II (adaptive attacks)  (f) Config-III (adaptive attacks)

(g) Config-IV  (h) Config-V  (i) Config-VI

Figure 9: Reconstruction results for CLIP-ViT-B/16 under various defense configurations.

Table 7: Performance of the optimization-based attack against various defenses, split at layer 12.

| Config | Method | CLIP-ViT-B/16 | | | DINOv2-Base | | |
|---|---|---|---|---|---|---|---|
| | | MS-SSIM ↑ | LPIPS ↓ | DINO ↑ | MS-SSIM ↑ | LPIPS ↓ | DINO ↑ |
| Config-I | rMLE | 0.1957 | 0.7881 | 0.1768 | 0.3840 | 0.5993 | 0.4710 |
| | LM | 0.1749 | 0.6690 | 0.1965 | 0.3953 | 0.5112 | 0.7031 |
| | GLASS | 0.2138 | 0.5568 | 0.4783 | 0.3881 | 0.4222 | 0.6667 |
| | DRAG | **0.3309** | **0.4793** | **0.6497** | **0.7392** | **0.1602** | **0.9370** |
| Config-II | rMLE | 0.1706 | 0.8145 | 0.1101 | 0.3840 | 0.5993 | 0.4710 |
| | LM | 0.1800 | 0.6838 | 0.1872 | 0.3953 | 0.5112 | 0.7031 |
| | GLASS | 0.2050 | 0.5627 | 0.4433 | 0.3881 | 0.4222 | 0.6667 |
| | DRAG | **0.2704** | **0.5264** | **0.6015** | **0.7392** | **0.1602** | **0.9370** |
| Config-III | rMLE | **0.1686** | 0.8128 | 0.1129 | 0.2469 | 0.6571 | 0.2463 |
| | LM | 0.1638 | 0.7079 | 0.1654 | 0.3198 | 0.5770 | 0.4781 |
| | GLASS | 0.1479 | **0.6225** | **0.3878** | 0.2699 | 0.4884 | 0.5310 |
| | DRAG | 0.0788 | 0.7449 | 0.3201 | **0.5703** | **0.2880** | **0.8432** |
| Config-I w/ adaptive | rMLE | 0.4062 | 0.5088 | 0.6880 | 0.4481 | 0.5796 | 0.5597 |
| | LM | 0.1765 | 0.6996 | 0.1962 | 0.4336 | 0.4916 | 0.8097 |
| | GLASS | 0.2321 | 0.5443 | 0.4770 | 0.4526 | 0.3849 | 0.6962 |
| | DRAG | **0.5930** | **0.2489** | **0.8772** | **0.7918** | **0.1218** | **0.9573** |
| Config-II w/ adaptive | rMLE | 0.3837 | 0.5498 | 0.6229 | 0.3851 | 0.6160 | 0.4820 |
| | LM | 0.1833 | 0.6816 | 0.2056 | 0.4155 | 0.4939 | 0.8124 |
| | GLASS | 0.2157 | 0.5630 | 0.4528 | 0.4242 | 0.3967 | 0.6954 |
| | DRAG | **0.6076** | **0.2347** | **0.8830** | **0.7783** | **0.1329** | **0.9489** |
| Config-III w/ adaptive | rMLE | 0.2101 | 0.7822 | 0.2072 | 0.3993 | 0.5794 | 0.5063 |
| | LM | 0.1696 | 0.6953 | 0.1818 | 0.4068 | 0.5104 | 0.7899 |
| | GLASS | 0.2402 | 0.5401 | 0.4862 | 0.4052 | 0.3971 | 0.6792 |
| | DRAG | **0.4557** | **0.3778** | **0.7557** | **0.7551** | **0.1467** | **0.9415** |
| Config-IV | rMLE | 0.2799 | 0.6604 | 0.4729 | 0.2428 | 0.6822 | 0.2546 |
| | LM | 0.1834 | 0.7159 | 0.2271 | 0.2852 | 0.6868 | 0.7270 |
| | GLASS | 0.1837 | 0.6707 | 0.3381 | 0.3499 | 0.4664 | 0.6782 |
| | DRAG | **0.5526** | **0.2925** | **0.8778** | **0.6252** | **0.2216** | **0.9243** |
| Config-V | rMLE | 0.2348 | 0.6939 | 0.3863 | 0.2306 | 0.6653 | 0.3183 |
| | LM | 0.1782 | 0.7377 | 0.2114 | 0.2546 | 0.7107 | 0.6272 |
| | GLASS | 0.1715 | 0.6864 | 0.3043 | 0.3710 | 0.4447 | 0.6446 |
| | DRAG | **0.4889** | **0.3369** | **0.8381** | **0.5235** | **0.2790** | **0.9106** |
| Config-VI | rMLE | 0.2270 | 0.7048 | 0.3747 | 0.2218 | 0.6821 | 0.3136 |
| | LM | 0.1783 | 0.7917 | 0.2099 | 0.2410 | 0.7140 | 0.5797 |
| | GLASS | 0.1950 | 0.7248 | 0.3141 | 0.3620 | 0.4339 | 0.6516 |
| | DRAG | **0.4469** | **0.3836** | **0.8096** | **0.5010** | **0.2977** | **0.9093** |

## A.3. Execution Time

The execution times for each attack algorithm are provided in Table 8.

Table 8: Execution times for optimization-based attacks at deepest split points.

| Method | # of Iterations | CLIP-ViT-B/16 | DINOv2-Base | CLIP-RN50 |
|---|---|---|---|---|
| rMLE | 20,000 | 6 min 42 s | 7 min 13 s | 6 min 04 s |
| LM | 20,000 | 24 min 20 s | 24 min 32 s | 22 min 27 s |
| GLASS (StyleGAN2-ADA-FFHQ) | 20,000 | 21 min 28 s | 22 min 02 s | 19 min 45 s |
| GLASS (StyleGAN-XL-ImageNet-1K) | 20,000 | 1 hr 37 min 08 s | 1 hr 37 min 02 s | 1 hr 34 min 34 s |
| DRAG (SDv1.5) | 4,000 | 32 min 52 s | 33 min 04 s | 32 min 40 s |

# B. Extended Experiments

## B.1. Scaling Reconstruction Schedule

Increasing $T$ or $k$ improves reconstruction performance by allowing more refinement steps, especially in the deeper layer. However, it also raises computational overhead. Figure 10a and Figure 10b visualizes the attack performance for different values of $T$ and $k$. Since $T = 250$ and $k = 16$ provides a satisfactory balance between performance and efficiency, we adopt it as the default hyperparameter setting.

## B.2. Guidance Strength $w$

The guidance strength $w$ balances feature matching and image prior during sampling. Increasing $w$ enhances guidance by focusing more on minimizing the distance $d_{\mathcal{H}}$, but this may compromise the realistic property $R_{\mathcal{I}}$ as defined in Equation (1). Conversely, an excessively low $w$ results in an unsuccessful attack due to insufficient guidance. Figure 10c presents the relationship between $w$ and reconstruction performance. We also tested Equation (8) with linear interpolation (lerp) and spherical linear interpolation (slerp), but found no significant performance differences.

## B.3. Importance of the Optimizer

Figure 10d compares the performance of attacks on IR from layer 12 with and without the Adam optimizer. The figure demonstrates that smoothing gradients with non-convex optimization techniques significantly enhances attack performance, especially at the deeper layer.

# C. Baseline Attacks

**rMLE.** (He et al., 2019) first proposed an optimization-based reconstruction attack that reconstructs $\mathbf{x}^*$ by optimizing a zero-initialized $\mathbf{x}$ to minimize $d_{\mathcal{H}}(f_c(\mathbf{x}), \mathbf{h}^*)$. To improve reconstruction quality, the method incorporates Total Variation regularization (Rudin et al., 1992) as an image prior:

$$\mathbf{x}' = \arg\min_{\mathbf{x} \in \mathcal{X}} d_{\mathcal{H}}(f_c(\mathbf{x}), \mathbf{h}^*) + \lambda_{\text{TV}} R_{\text{TV}}(\mathbf{x}). \tag{13}$$

**LM.** (Singh et al., 2021) enhances reconstruction quality by applying a deep image prior (Ulyanov et al., 2018) to regularize $\mathbf{x}$. Rather than updating $\mathbf{x}$, they re-parameterize it as the output of a CNN-based image synthesis model $f_\theta(\epsilon)$, where the fixed input $\epsilon \sim \mathcal{N}(\mathbf{0}, \mathbf{I})$ remain constant during optimization, while the model parameters $\theta$ are optimized:

$$\mathbf{x}' = \arg\min_{\theta} d_{\mathcal{H}}(f_c(f_\theta(\epsilon)), \mathbf{h}^*) + \lambda_{\text{TV}} R_{\text{TV}}(\mathbf{x}). \tag{14}$$

Since ViTs divide images into non-overlapping patches, directly optimizing $\mathbf{x}$ often yields visible artifacts at patch boundaries. To alleviate this, we add the patch-smoothness prior $R_{\text{patch}}$ from Hatamizadeh et al. (2022) to both the rMLE and LM

objectives when reconstructing images from ViTs:

$$R_{\text{patch}}(\mathbf{x}) = \sum_{k=1}^{\frac{H}{P}-1} \|\mathbf{x}[:, P \cdot k, :, :] - \mathbf{x}[:, P \cdot k - 1, :, :]\|_2 + \sum_{k=1}^{\frac{W}{P}-1} \|\mathbf{x}[:, :, P \cdot k, :] - \mathbf{x}[:, :, P \cdot k - 1, :]\|_2, \tag{15}$$

where $P$ denotes the patch size of the ViT model.

**GLASS.** (Li et al., 2023) proposed a scenario in which the adversary has knowledge of the data distribution and access to auxiliary data for training a StyleGAN. Instead of directly updating $\mathbf{x}$, the adversary updates the latent code $\mathbf{z} \in \mathcal{Z}$ or the style code $\mathbf{w}^+ \in \mathcal{W}^+$ to improve the quality of the generated image. In the first stage, the latent code $\mathbf{z}$ is randomly

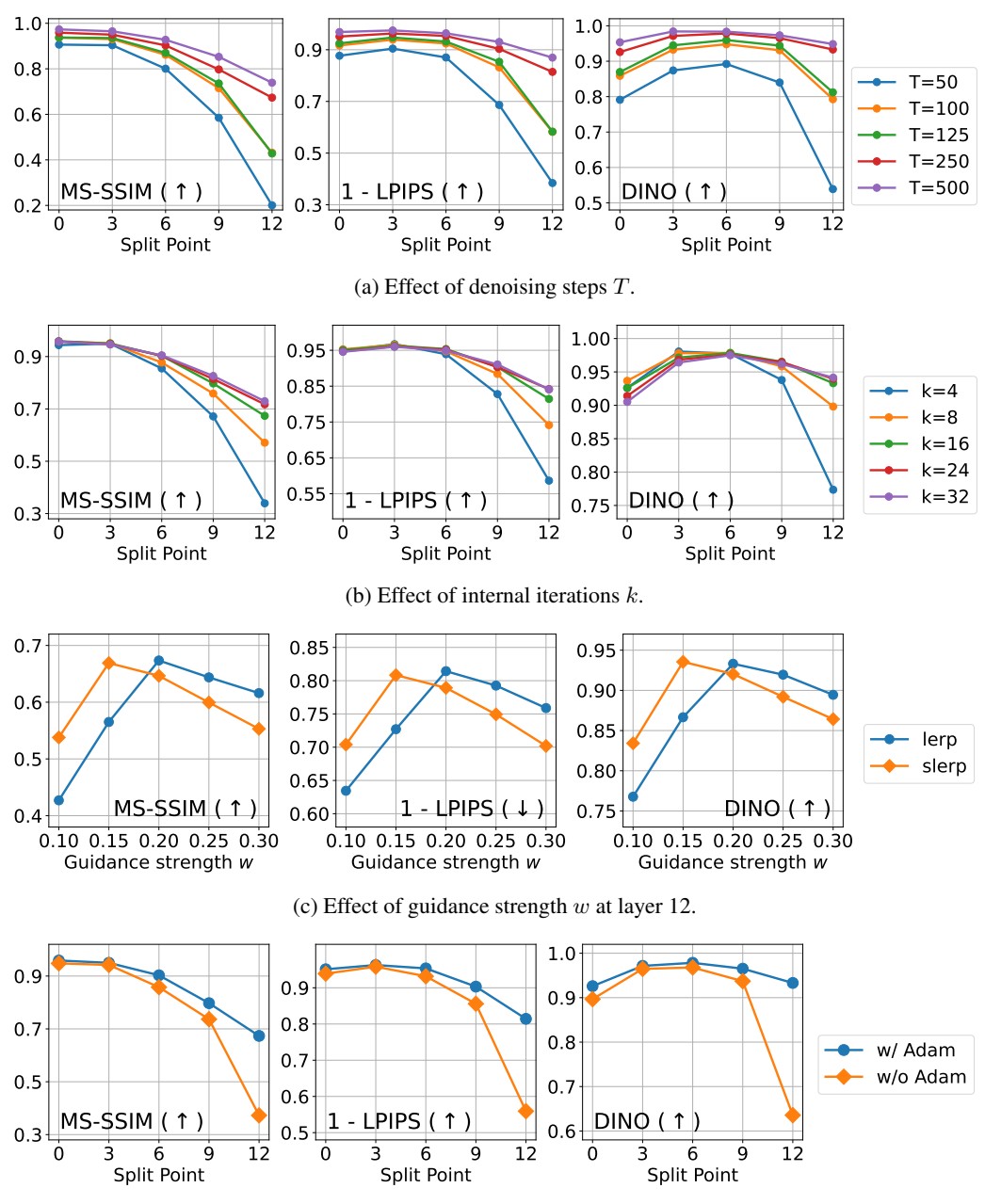

(a) Effect of denoising steps $T$.

(b) Effect of internal iterations $k$.

(c) Effect of guidance strength $w$ at layer 12.

(d) Performance comparison of DRAG with and without the Adam optimizer.

Figure 10: Hyperparameter sensitivity analysis for DRAG on CLIP-ViT-B/16.

initialized from a standard normal distribution, $\mathbf{z} \sim \mathcal{N}(\mathbf{0}, \mathbf{I})$, and is then update as follow:

$$\mathbf{z}' = \arg \min_{\mathbf{z} \in \mathcal{Z}} d_{\mathcal{H}}(f_c(G(f_{\mathrm{map}}(\mathbf{z}))), \mathbf{h}^*) + \lambda_{\mathrm{TV}} R_{\mathrm{TV}}(\mathbf{x}) + \lambda_{\mathrm{KL}} R_{\mathrm{KL}}(\mathbf{z}), \tag{16}$$

where $R_{\mathrm{KL}}$ is the Kullback-Leibler divergence that regularize the latent code $\mathbf{z}$. The updated latent code $\mathbf{z}'$ is then transformed to the style code $\mathbf{w}^+ = f_{\mathrm{map}}(\mathbf{z}')$, which is subsequently updated for fine-grained reconstruction:

$$\mathbf{w}^+ = \arg \min_{\mathbf{w}^+} d_{\mathcal{H}}(f_c(G(\mathbf{w}^+)), \mathbf{h}^*) + \lambda_{\mathrm{TV}} R_{\mathrm{TV}}(\mathbf{x}). \tag{17}$$

For class-conditioned GANs, we further optimize a zero-initialized class logits vector $\mathbf{l} \in \mathbb{R}^{1000}$, which is normalized via the softmax function to produce class probabilities $\mathbf{p} = \mathrm{softmax}(\mathbf{l})$ during the forward pass $\mathbf{w}^+ = f_{\mathrm{map}}(\mathbf{z}', \mathbf{Ep})$. Then $\mathbf{p}$ are multiplied by the class embeddings $\mathbf{E} \in \mathbb{R}^{d \times 1000}$ to compute the class-specific latent code $\mathbf{Ep}$.

## D. Defensive Algorithms

### D.1. Privacy Leakage Mitigation Methods

**DISCO.** (Singh et al., 2021) introduces a method to mitigate privacy leakage by pruning a subset of the IRs' channels before transmitting them to the server. Specifically, the pruning operation $\mathbf{h}' = f_p(\mathbf{h}, r_p)$ is performed using an auxiliary channel pruning module $f_p$, where the pruning ratio $r_p$ controls the proportion of pruned channels. This ratio can be dynamically adjusted during model inference. The pruning module $f_p$ is trained in a min-max framework, where $f_p$ minimizes privacy leakage by maximizing the reconstruction loss, and the inverse network $f_c^{-1}$ minimizes the reconstruction loss:

$$\begin{aligned} L_{\mathrm{util}} &= \mathbb{E}[\ell_{\mathrm{util}}(f_s(\mathbf{h}'), y)], \\ L_{\mathrm{privacy}} &= \mathbb{E}[||f_c^{-1}(\mathbf{h}') - \mathbf{x}||_2], \\ &\min_{f_p} [\max_{f_c^{-1}} -L_{\mathrm{privacy}} + \rho_D \min_{f_c, f_s} L_{\mathrm{util}}]. \end{aligned} \tag{18}$$

**NoPeek.** (Vepakomma et al., 2020) aims to mitigate privacy leakage by training models to minimize the mutual information $I(\mathbf{X}; \mathbf{H})$ between the input data $\mathbf{X}$ and the intermediate representation $\mathbf{H}$. Since directly calculating $I(\mathbf{X}; \mathbf{H})$ is challenging, the authors propose using distance correlation (dCor) as a surrogate measure:

$$\min_{f_c, f_s} \mathbb{E}[\rho_N \cdot \mathrm{dCor}(f_c(\mathbf{x}), \mathbf{x}) + \ell_{\mathrm{util}}(f_s(f_c(\mathbf{x})), y)]. \tag{19}$$

While Vepakomma et al. (2020) assumes that users pre-train the target models $f$ from scratch using distance correlation loss, our experiments differ by applying the loss during model fine-tuning. This adaptation allows us to leverage pre-existing model knowledge while still addressing privacy concerns.

### D.2. Implementation of Defense Mechanisms

Since the target models are not pre-trained on ImageNet-1K, they lack classification heads tailored for the ImageNet-1K classification task. While it is possible to directly initialize random classification heads and train with DISCO or NoPeek, this approach significantly degrades accuracy. To address this, we prepare target models $f$ through a two-stage process. First, we perform linear probing by freezing the pre-trained backbone and training only the classification head on $D_{\mathrm{private}}$. Then, we fine-tune the entire model using the selected defense mechanism. This strategy helps preserve classification performance during the defensive training phase.

To ensure the effectiveness and robustness of DRAG, we adopt an informed defender threat model, assuming the client has full knowledge of DRAG. Under this assumption, defenders can design countermeasures by leveraging insights from the methodology of DRAG. Specifically, for DISCO, we employ an inverse network with the same architecture as described in Section 4.4. For NoPeek, we adopt the same distance metric $d_{\mathcal{H}}$ as defined in Equation (11), which reflects how an informed defender would calibrate a privacy-preserving model against DRAG.

# E. Implementation Details

We list the hyperparameters for various optimization-based and learning-based reconstruction attacks in Table 9 and Table 10, respectively. The experiments were conducted on a server equipped with 384 GB RAM, two Intel Xeon Gold 6226R CPUs, and eight NVIDIA RTX A6000 GPUs.

The implementation of rMLE (He et al., 2019), LM (Singh et al., 2021), DISCO (Singh et al., 2021) and NoPeek (Vepakomma et al., 2020) are adapted from prior works.[1]

Table 9: Default hyperparameters for the optimization-based reconstruction attacks.

|  | rMLE | LM | GLASS |
|---|---|---|---|
| Optimizer | Adam (lr = 0.05) | Adam (lr = 0.01) | Adam (lr = 0.01) |
| Num of iters ($n$) | 20,000 | 20,000 | 20,000 |
| Pretrained model | - | - | StyleGAN2-ADA (FFHQ) |
|  |  |  | StyleGAN-XL (ImageNet-1K) |
| $\lambda_{TV}$ | 1.5 | 0.05 | 0 |
| $\lambda_{patch}$ | 0.001 | 0.001 | 0 |
| $\lambda_{\ell_2}$ | 0 | 0 | 0 |
| $\lambda_{KL}$ | - | - | 1.0 |
|  | DRAG | DRAG++ | |
| Strength ($s$) | 1.0 | 0.3 | |
| DDIM randomness ($\eta$) | 1.0 | 1.0 | |
| Guidance strength ($w$) | 0.2 | 0.2 | |
| Max grad norm ($c_{\max}$) | 0.02 | 0.02 | |
| Sampling steps ($T$) | 250 | 250 | |
| Self-recurrence ($k$) | 16 | 16 | |
| $\lambda_{TV}$ | 0 | 0 | |
| $\lambda_{patch}$ | 0 | 0 | |
| $\lambda_{\ell_2}$ | 0.01 | 0.01 | |

Table 10: Hyperparameters for inverse network training across all experiments.

| | |
|---|---|
| Optimizer | Adam (lr = 0.001) |
| LR Scheduler | Cosine annealing (linear warm-up in 5000 iterations) |
| Number of iterations | 100,000 |
| Batch size | 256 |
| Mask ratio ($r_{mask}$) | 0.25 |

---

[1]https://github.com/aidecentralized/InferenceBenchmark

