# OpenReview forum: "DRAG: Data Reconstruction Attack using Guided Diffusion"
_ICML.cc/2025/Conference — ICML 2025 poster_

### Official Review · Reviewer_i33r · 2025-03-13

**Overall Recommendation:** 2

**Summary:**

This paper proposes DRAG, a new data reconstruction attack under the guidance of diffusion models. This method utilizes the rich prior knowledge embedded in the latent diffusion model and firstly reconstructs data from vision foundation models. Experiments have shown the superiority of DRAG to some extent.

**Claims And Evidence:**

No, the experiments are insufficient and certain claims need further justifications. Please refer to the later parts of this review for details.

**Essential References Not Discussed:**

Yes, there are certain new works [1-4] that are not evaluated in this paper.

**Experimental Designs Or Analyses:**

The experimental results in this paper are insufficient and problematic. Detailed comments are listed as follows:
- Current evaluations mainly focus on the image fidelity of reconstructed images without fully considering the potential privacy threat for users. More discussion of the metrics are expected.
- The compared baselines are not sufficient. It would be better to provide comparisons with more SOTA attacks such as [1-4]. The listed methods are also evaluated in the previous data reconstruction attacks.
- The assumption that the attacker has white-box access to the model architecture and parameters is a strong setting, which is usually unpractical in real-world scenarios. However, previous works [4-5] have discussed the utility of their methods under black-box scenarios. More discussion on the more practical settings are expected.
- What about the performance on smaller CNN models like ResNet? More evaluation on the CNN models utilized in previous works is expected for alignment with the baselines.

[1]Dario Pasquini, Giuseppe Ateniese, and Massimo Bernaschi. Unleashing the tiger: Inference attacks on split learning. In Proceedings of the 2021 ACM SIGSAC Conference on Computer and Communications Security, pages 2113–2129, 2021.
[2]Unsplit: Data-oblivious model inversion, model stealing, and label inference attacks against split learning. In Proceedings of the 21st Workshop on Privacy in the Electronic Society.
[3]Xinben Gao and Lan Zhang. PCAT: Functionality and data stealing from split learning by Pseudo-Client attack. In 32nd USENIX Security Symposium (USENIX Security 23), pages 5271–5288, Anaheim, CA, 2023. USENIX Association.
[4]Xu X, Yang M, Yi W, et al. A Stealthy Wrongdoer: Feature-Oriented Reconstruction Attack against Split Learning[C]//Proceedings of the IEEE/CVF Conference on Computer Vision and Pattern Recognition. 2024: 12130-12139.
[5]Li Z, Yang M, Liu Y, et al. GAN you see me? enhanced data reconstruction attacks against split inference[J]. Advances in Neural Information Processing Systems, 2023, 36: 54554-54566.

**Methods And Evaluation Criteria:**

Yes, this method is the first data reconstruction attack that introduces the diffusion model.

**Other Comments Or Suggestions:**

- minor mistake: the GLASS [5] is published in 2023 instead of 2024.

**Other Strengths And Weaknesses:**

Other Strengths:
- The proposed method is not time-consuming.

Other Weaknesses:
- The Peak Signal-to-Noise Ratio (PSNR) metric is also critical to assess the image fidelity. However, this paper does not adopt this metric.

**Questions For Authors:**

- The GradViT is introduced in the “Baseline Attacks” section in the Appendix. But where is the experimental results?

**Relation To Broader Scientific Literature:**

This paper focuses on an important privacy problem. However, it relies on a strong setting of white-box access to the target model. This will limit its potential impact on broader scientific literature.

**Theoretical Claims:**

Yes, the theoretical claims are correct.

---

> ### Author Rebuttal · Authors · 2025-04-01
>
> Thank you for your valuable feedback. We address your concerns point by point below.
>
> ---
> > 1. Related to the evaluation metrics
>
> The choice of metrics is highly application dependent, and our selections were guided by prior works in this area. In our study, we focused on MS-SSIM, LPIPS, and DINO because they better capture human perceptual similarity and privacy leaks. PSNR and MSE are sensitive to translation and other low-level distortions. However, we are open to including PSNR to offer a more comprehensive analysis.
>
> ---
> > 2. It would be better to provide comparisons with more SOTA attacks.
> > 3. The assumption that the attacker has white-box access to the model architecture and parameters is a strong setting, which is usually unpractical in real-world scenarios.
> >
> We agree that FORA [1] and related works are important to discuss in the context of privacy threats in split inference, and we will include a discussion of these methods in the revised paper.
>
> These works consider privacy risk under a different configuration from ours by exploring query-free data reconstruction attacks in split learning. In that setup, the attacker cannot directly access $f_c$ but may capture or interfere with the training process to build a surrogate model $\tilde{f}_c \approx f_c$. Once this surrogate model is built, attackers can reconstruct private data using either optimization-based (e.g., DRAG) or learning-based methods. Xu et al. [1] note that combining these two research areas can lead to more powerful reconstruction attacks, as their developments are independent.
>
> On the other hand, our work focuses on the privacy risks associated with using foundation models as part of the model parameters in downstream tasks, implying that an attacker can feasibly access $f_c$ directly. Our findings highlight the need to develop privacy-preserving inference techniques, especially as new applications [2, 3] increasingly leverage foundation models.
>
> ---
> > 4. What about the performance on smaller CNN models like ResNet? More evaluation on the CNN models utilized in previous works is expected for alignment with the baselines.
>
> DRAG is broadly applicable to all models regardless of architecture. To address it, we evaluated our method on CLIP-RN50. Key results are presented in Table 2 of the main paper. Detailed experimental results can be found in Appendix A, specifically Table 6, Fig. 6c) and Fig. 7c). We will reorganize the paper to more clearly direct readers to the experimental results and improve overall clarity.
>
> The table below—captured from Appendix A—illustrates DRAG's effectiveness compared to other methods in reconstructing data from CLIP-RN50. In this experiment, the feature space distance metric $d_\mathcal{H}$ was implemented using MSELoss. Due to space limitations, we have presented results for model splits at blocks 4 to 5.
>
> | Split Point | Method | MS-SSIM ($\uparrow$) | LPIPS ($\downarrow$) | DINO ($\uparrow$) |
> | - | - |:-:|:-:|:-:|
> |Block 4|rMLE |  0.4888  |  0.4198  |  0.7776  |
> |       |LM   |  0.5855  |  0.2576  |  0.9012  |
> |       |GLASS|  0.4872  |  0.3568  |  0.7315  |
> |       |DRAG |**0.7896**|**0.0898**|**0.9622**|
> |Block 5|rMLE |  0.3980  |  0.5006  |  0.6739  |
> |       |LM   |  0.4432  |  0.3409  |  0.7614  |
> |       |GLASS|  0.2917  |  0.4223  |  0.6811  |
> |       |DRAG |**0.5206**|**0.2231**|**0.9001**|
>
> ---
> > Minor mistake: the GLASS is published in 2023 instead of 2024.
>
> Thanks for pointing out this issue. We have revised the year of this reference.
>
> ---
> > The GradViT is introduced in the “Baseline Attacks” section in the Appendix. But where is the experimental results?
>
> We apologize for the confusion regarding the explanation of GradViT. GradViT is a parallel work that focuses on reconstructing training data using the gradients of model parameters. We referenced it because we observed artifacts in rMLE when attacking ViT, and GradViT proposed a regularization to mitigate these artifacts. We have adapted this regularization to strengthen rMLE and LM (referring to $\lambda_\text{patch}$ in Table 10). However, even with this regularization, reconstruction of deep-layer IR fails, which motivated our proposal of a data-driven image prior for enhancement.
>
> ---
> We appreciate your feedback and remain available to address any additional questions or concerns you may have.
>
> ### Reference
> [1] Xu, X., et al. A stealthy wrongdoer: Feature-oriented reconstruction attack against split learning. In CVPR, 2024.
>
> [2] Liu, H., et al. Visual instruction tuning. In NeurIPS, 2023.
>
> [3] Chen, J., et al. Minigpt-v2: large language model as a unified interface for vision-language multi-task learning. arXiv preprint arXiv:2310.09478. 2023

---

### Official Review · Reviewer_WYvj · 2025-03-13

**Overall Recommendation:** 2

**Summary:**

- This paper is about reconstruction attacks in split inference (SI) configurations. Specifically, this paper studies reconstructing a datapoint given the intermediate representation of that datapoint in a deep models
- The paper proposed guided diffusion to do this attack (DRAG), where the guidance term is given by a cosine similarity between the reconstruction intermediate embedding and the target embedding
- The authors validate they method on pretrained CLIP models, and for models with defences applied, showing good results compared to existing methods
- DRAG++ is also proposed (in a very rushed fashion) which also uses an inverse model to bootstrap the attack

**Claims And Evidence:**

Most of the claims are substantiated. See other sections for specific details.

I think that generally speaking, the attack seems promising and the results are convincing, particularly when compared to equivalent attacks in the same settings. However, it is unclear what dataset the GAN used in GLASS is trained on, and how that compared to the dataset that the diffusion model DRAG is trained on, and should be mentioned in more detail

**Essential References Not Discussed:**

None that I am aware of.

**Experimental Designs Or Analyses:**

There are a few concerns:
1. The largest concern is that the paper is that there is both the reconstruction model and the attack diffusion model are trained on the same domain. Appendix A.3. claims that they do this, testing on the UCMerced LandUse data, but because they are using pretrained diffusion models (stable Diffusion 1.5), it is unclear how much contamination there is in the diffusion model.
2. I am not convinced that fine-tuning the bast model on distinct subsets of the dataset ensures separation, as the base model is already pretrained on data which might include both the private and public splits of data.
3. To fix this, I think it would be best to train models from scratch (at least in one section of the paper). In this case the inference model could, for example, be trained on CelebA, and the diffusion model be trained on ImageNet. This would ensure that there is no leakage in the paper.

**Methods And Evaluation Criteria:**

There is very limited explanation of the DRAG++ method. In particular:
1. Can we write out the DRAG++ pseudocode in full? At least in the appendix
2. What are the details of the dataset used to train the reconstruction network? How sensitive is DRAG++ to the choice of public dataset
3. I think DRAG++ should be explained earlier in the paper rather than at the end right before the conclusion. At the moment it seems like it was added ad-hoc

**Other Comments Or Suggestions:**

None.

**Other Strengths And Weaknesses:**

Main concerns are in the experimental design/analysis section.

**Questions For Authors:**

- For ResNet models, do you still use the same cosine loss function given in equation 10? The authors only specify this for transformer models.

**Relation To Broader Scientific Literature:**

This paper proposes another reconstruction attack for the split inference setting. It is not particularly surprising that such an attack works, however this is good validation. I think that using diffusion models in reconstruction attacks is a promising general research direction.

**Theoretical Claims:**

There are no theoretical concerns in this paper.

---

> ### Author Rebuttal · Authors · 2025-04-01
>
> Thank you for your valuable feedback. We address your concerns point by point below.
>
> ---
> > it is unclear what dataset the GAN used in GLASS is trained on, and how that compared to the dataset that the diffusion model DRAG is trained on, and should be mentioned in more detail.
>
> In our evaluation of GLASS, we used two GAN models: StyleGAN2-ADA (trained on FFHQ) and StyleGAN-XL (trained on ImageNet). These models are the most relevant publicly accessible GANs, to the best of our knowledge. We assume that GLASS has prior knowledge of the target image distribution; therefore, if the private image is from FFHQ, the attacker selects StyleGAN2-ADA, and if not, StyleGAN-XL is selected. This setup inherently gives advantages to GLASS in the comparison.
>
> On the other hand, DRAG utilizes SDv1.5, pretrained on a subset of LAION-5B, to demonstrate the effectiveness of the diffusion prior with the support of a large dataset. This choice reflects the evolving nature of DMs and their accessibility, potentially reducing the attacker's cost in preparing the model. We will update the paper to include these experimental details for improved clarity.
>
> ---
> > There is very limited explanation of the DRAG++ method. In particular:
> > 1. Can we write out the DRAG++ pseudocode in full? At least in the appendix
> > 2. What are the details of the dataset used to train the reconstruction network? How sensitive is DRAG++ to the choice of public dataset
> > 3. I think DRAG++ should be explained earlier in the paper rather than at the end right before the conclusion. At the moment it seems like it was added ad-hoc
>
> We agree that DRAG++ should be introduced earlier in the paper. We will also include the full DRAG++ pseudocode in the appendix to provide complete details. In brief, DRAG++ uses an auxiliary $f_c^{-1}$ to initialize $x_{t}$ and denoises from $t=sT$ (where $s \in [0,1]$), while the core guided diffusion remains unchanged. We refer to DRAG++ as an optional enhancement, since $f_c^{-1}$ is an auxiliary component for attackers who have the resources to train such a network. Regarding the training dataset of $f_c^{-1}$, we train it on the ImageNet-1K training split (using 50% of the data, while assuming the other 50% is not accessible to the attacker).
>
> To evaluate the sensitivity of DRAG++ to the choice of public dataset, we also train $f_c^{-1}$ on other datasets. For a fair comparison, we randomly sampled 60,000 images from the dataset to serve as the training data. Our results indicate that using a less diverse dataset (e.g., FFHQ) leads to a slight performance drop, whereas training on more complex datasets (e.g., ImageNet or MSCOCO), maintains similar performance.
>
> | Split Point | Dataset | MS-SSIM ($\uparrow$) | LPIPS ($\downarrow$) | DINO ($\uparrow$) |
> |--|:- |:-:|:-:|:-:|
> |Layer 9| ImageNet |0.8062|0.0914|0.9682|
> || MSCOCO   |0.8037|0.0944|0.9655|
> || FFHQ     |0.7805|0.1021|0.9632|
> |Layer 12| ImageNet |0.6987|0.1732|0.9412|
> || MSCOCO   |0.6850|0.1867|0.9407|
> || FFHQ     |0.6568|0.2092|0.9325|
>
> ---
> > The largest concern is that the paper is that there is both the reconstruction model and the attack DM are trained on the same domain.
>
> To address this concern, we designed an experiment to evaluate the OOD capability of using a DM as an image prior in DRA. In this experiment, we employ the checkpoint "google/ddpm-bedroom-256" from HuggingFace, which we denote as DRAG* to distinguish it from DRAG using SDv1.5. The inference model used is CLIP-ViT-B/16, and the evaluation is performed on the same dataset as in our paper. The bold number, shown among rMLE, LM, GLASS, and DRAG*, indicates the best score. The original score of DRAG is also provided for reference.
>
> | Split Point | Method | MS-SSIM ($\uparrow$) | LPIPS ($\downarrow$) | DINO ($\uparrow$) |
> |:-|:-|:-:|:-:|:-:|
> | Layer 9 | rMLE | 0.4957    | 0.5131    | 0.7086    |
> || LM   | **0.6681**| **0.2138**| **0.9037**|
> || GLASS| 0.3852    | 0.4310    | 0.6648    |
> || DRAG*|   0.5378  |   0.3940  |   0.8147  |
> || DRAG |   0.7974  |   0.0967  |   0.9652  |
> | Layer 12| rMLE |   0.3884  | 0.5900    |   0.6462  |
> || LM   | 0.2560    | 0.6024    | 0.4097    |
> || GLASS| 0.2396    |   0.5790  | 0.4578    |
> || DRAG*| **0.3958**| **0.4941**| **0.7240**|
> || DRAG |   0.6735  |   0.1857  |   0.9331  |
>
> These experimental results show that DRAG using an OOD DM outperforms rMLE, LM, and GLASS in Layer 12 according to LPIPS and DINO metrics, thereby demonstrating the effectiveness of leveraging DMs in DRAs under an OOD configuration.
>
> ---
> > For ResNet models, do you still use the same cosine loss function given in equation 10? The authors only specify this for transformer models.
>
> We have applied MSELoss as the distance metric $d_\mathcal{H}$ for all DRAs while attacking CLIP-RN50. We would mention this configuration earlier in the revised menuscript.
>
> ---
> We appreciate your feedback and remain available to address any additional questions or concerns you may have.

---

### Official Review · Reviewer_DMzW · 2025-03-14

**Overall Recommendation:** 2

**Summary:**

This paper proposes a data reconstruction attack in split inference. The proposed method is based on guided diffusion, which leverages the rich prior knowledge embedded in a latent diffusion model (LDM) pre-trained on a large-scale dataset. The proposed method performs iterative reconstruction on the LDM’s learned image prior, effectively generating high-fidelity images resembling the original data from their intermediate representations (IR). Extensive experiments demonstrate that the proposed approach outperforms prior methods.

**Claims And Evidence:**

I didn't find claims that are problematic.

**Essential References Not Discussed:**

Below is a work solving the same problem by using diffusion model, and the target model is ViT. I suggest comparing DRAG with this work.

Chen, D., Li, S., Zhang, Y., Li, C., Kundu, S. and Beerel, P.A., 2024. DIA: Diffusion based Inverse Network Attack on Collaborative Inference. In Proceedings of the IEEE/CVF Conference on Computer Vision and Pattern Recognition (pp. 124-130).

**Experimental Designs Or Analyses:**

This paper lacks experimental details. There is no information is provided for the number of timestep T, the t in Algorithm 1.

Besides the above parameters, the experiments are comprehensive and convincing.

**Methods And Evaluation Criteria:**

The proposed method make sense for the problem and application. However, the paper lacks details on the attack framework. The author only refer to the Figure 2 for the attack framework. In Figure 2, why all images look like noise for timesteps 0 to T? g_t in equation 6 is not clear shown in the figure. Algorithm 1 is never cited, and the symbols in the algorithm are not explained. In line 225, it is unclear why there is a loop and what is the value for k in the experiments. If the attack requires back propogation for every timestep, the attack should experience very long runtime. If the attack does not require back propogation for every timestep, it is unclear what backprop. is done for the diffusion model.

The evaluation criteria is solid.

**Other Comments Or Suggestions:**

I don't have additional comments or suggestions.

**Other Strengths And Weaknesses:**

Strengths:

This paper provides defense methods to DRA.

Weakness:

In section 2.2 line 98-99, the author mentions three types of DRA, however, only one of them is introduced in this section.

**Questions For Authors:**

How does the back propagation applied to the diffusion model as suggested in Figure 2?

**Relation To Broader Scientific Literature:**

This work focus on an important research area. However, this paper lacks compare with recent methods which addressing the same proble. See below.

**Theoretical Claims:**

There is no theoretical claim and proof.

---

> ### Author Rebuttal · Authors · 2025-04-01
>
> Thank you for your valuable feedback. We address your concerns point by point below.
>
> ---
> > The proposed method make sense for the problem and application. However, the paper lacks details on the attack framework. The author only refer to the Figure 2 for the attack framework. In Figure 2, why all images look like noise for timesteps 0 to $T$? $g_t$ in equation 6 is not clear shown in the figure. Algorithm 1 is never cited, and the symbols in the algorithm are not explained.
>
> We apologize for the lack of clarity regarding the DRAG framework in the draft. In the revised manuscript, we will provide a detailed description of the DRAG approach and address several points. In Figure 2, we currently want to decipt that $x_{t-1}$ to $x_0$ are to be computed, and we will further improve the legend to enhance clarity. Additionally, we will explicitly illustrate and define $g_t$ in Figure 2 to clearly depict its role. We will also properly reference Alg. 1 in the main paper and include thorough explanations for all associated symbols.
>
> ---
> > In line 225, it is unclear why there is a loop and what is the value for $k$ in the experiments. If the attack requires back propogation for every timestep, the attack should experience very long runtime. If the attack does not require back propogation for every timestep, it is unclear what backprop. is done for the diffusion model.
> >
> > How does the back propagation applied to the diffusion model as suggested in Figure 2?
>
> To clarify, DRAG requires back-propagation at each timestep to reconstruct the image, and the number of iterations $k$ in the inner loop controls the reconstruction quality. Figure 11b) illustrates the trade-off between reconstruction quality and execution time, while Table 9 provides execution times for various DRAs. Although the algorithm includes an inner loop, our experiments demonstrate that this approach leads to higher reconstruction performance when the split point is deep, thereby highlighting significant privacy threats.
>
> The back propagation computes the gradient for the sample $x_t$, which is then used to adjust the sampling process of $\epsilon_t$ according to Eq. 8 during DDPM denoising.
>
> ---
> > In section 2.2 line 98-99, the author mentions three types of DRA, however, only one of them is introduced in this section.
>
> Thanks for pointing out this issue. We will revise this section to mention the other types of DRA for a more comprehensive review.
>
> ---
> > Below is a work solving the same problem by using diffusion model, and the target model is ViT. I suggest comparing DRAG with DIA.
>
> In our framework, DIA can be viewed as an enhancement to the inverse network $f_c^{-1}$ within our reconstruction pipeline (see Fig. 2), as it provides a better initialization for the optimization-based reconstruction process. Notably, the $f_c^{-1}$ component is optional and requires extra data and model training. In contrast, DRAG leverages a publicly available diffusion model and does not require additional training or data, aligning with the prior work we compared against.
>
> ---
> We appreciate your feedback and remain available to address any additional questions or concerns you may have.

---

### Official Review · Reviewer_u9u9 · 2025-03-14

**Overall Recommendation:** 3

**Summary:**

The paper introduces a new reconstruction attack method, DRAG (Data Reconstruction Attack using Guided Diffusion), that reconstructs private data from intermediate representations in split Inference settings. Unlike previous attacks on small CNNs, DRAG employs Latent Diffusion Models (LDMs) to iteratively improve reconstructions, resulting in high-fidelity image recovery from the deep-layer intermediate representations of CLIP and DINOv2. The experimental results indicate that the propsoed method surpasses existing attacks (e.g., rMLE, LM, GLASS) in deep layers and maintains effectiveness against privacy defenses such as DISCO and NoPeek. Furthermore, the enhanced version, DRAG++, utilizes an inverse network for improved initialization, leading to higher attack success rates. The results highlight significant privacy risks in vision foundation models, emphasizing the necessity for stronger defenses within SI frameworks.

**Claims And Evidence:**

The paper presents empirical evidence supporting its claims through quantitative experiments on several benchmarks (MSCOCO, FFHQ, ImageNet-1K) and thorough comparisons with previous data reconstruction attacks (rMLE, LM, GLASS). The findings indicate that DRAG offers superior reconstruction quality, especially at deeper layers of CLIP and DINOv2. This reinforces the main assertion that large vision foundation models are susceptible to privacy attacks in split inference (SI) scenarios.

**Essential References Not Discussed:**

A Stealthy Wrongdoer: Feature-Oriented Reconstruction Attack against Split Learning, CVPR 2024
GAN-based data reconstruction attacks in split learning,

**Experimental Designs Or Analyses:**

The paper shows a comprehensive experimental design that evaluates prominent vision models (CLIP, DINOv2) and benchmark datasets (MSCOCO, FFHQ, ImageNet-1K). It effectively compares previous attack methods (rMLE, LM, GLASS) and incorporates multiple reconstruction quality metrics (MS-SSIM, LPIPS, DINO similarity) for an in-depth analysis. However, The paper evaluates DRAG against DISCO (2021) and NoPeek (2020), which are relatively older defenses in the evolving landscape of privacy-preserving machine learning.

**Methods And Evaluation Criteria:**

The methods and evaluation criteria are well-suited for assessing reconstruction quality, utilizing vision foundation models such as CLIP and DINOv2 alongside benchmark datasets like MSCOCO, FFHQ, and ImageNet-1K. Metrics including MS-SSIM, LPIPS, and DINO similarity effectively evaluate both low- and high-level fidelity. Additionally, comparisons with rMLE, LM, and GLASS confirm the improvements made. The inclusion of privacy defenses (DISCO and NoPeek) further strengthens the analysis.

**Other Comments Or Suggestions:**

Please refer to Other Strengths and Weaknesses.

**Other Strengths And Weaknesses:**

Weakness:


+ Figure 2 is not very effective for understanding, even though it is drawn simply. The caption should include additional explanations.

+ The paper evaluates DRAG against DISCO (2021) and NoPeek (2020), but these defenses are somewhat outdated.

+ The method is not specifically designed for ViT or deeper layer models. Its applicability to CNNs and potential performance improvements over other methods remain unclear.

+ The experimental results lack discussion. The method is more effective at reconstructing data from deep layers compared to shallow layers. What accounts for this difference?

+ The paper should include the discussion of FORA (A Stealthy Wrongdoer: Feature-Oriented Reconstruction Attack against Split Learning 2024 CVPR), etc,. methods.

**Questions For Authors:**

Please refer to Other Strengths and Weaknesses.

**Relation To Broader Scientific Literature:**

The paper builds upon prior work in **data reconstruction attacks**, **diffusion models**, and **split inference** privacy risks, contributing a novel diffusion-guided approach for reconstructing private data from intermediate representations.

**Theoretical Claims:**

This paper is primarily empirical rather than theoretical, focusing on experimental validation of data reconstruction attacks rather than formal proofs.

---

> ### Author Rebuttal · Authors · 2025-04-01
>
> Thank you for your valuable feedback. We address your concerns point by point below.
>
> ---
> > Fig. 2 is not very effective for understanding, even though it is drawn simply. The caption should include additional explanations.
>
> We agree that enhancing the caption and associated text will improve clarity. We will revise the figure accordingly and upload the updated version to OpenReview before 4/8.
>
> ---
> > The paper evaluates DRAG against DISCO (2021) and NoPeek (2020), but these defenses are somewhat outdated.
>
> Thank you for pointing out this issue. We chose DISCO and NoPeek as target defenses because they are representative of the approaches highlighted in GLASS, and they provide a well-established baseline for comparison. Additionally, we have identified several more recent works [4-6], and we will include evaluations against these newer defenses in the next revision to offer a more comprehensive assessment.
>
> ---
> > The method is not specifically designed for ViT or deeper layer models. Its applicability to CNNs and potential performance improvements over other methods remain unclear.
>
> DRAG is broadly applicable to all models regardless of architecture. To address it, we evaluated our method on CLIP-RN50. Key results are presented in Table 2 of the main paper. Detailed experimental results can be found in Appendix A, specifically Table 6, Fig. 6c) and Fig. 7c). We will reorganize the paper to more clearly direct readers to the experimental results and improve overall clarity.
>
> The table below—captured from Appendix A—illustrates DRAG's effectiveness compared to other methods in reconstructing data from CLIP-RN50. In this experiment, the $d_\mathcal{H}$ was implemented using MSELoss. Due to space limitations, we have presented results for model splits at blocks 4 to 5.
>
> |Split Point| Method | MS-SSIM ($\uparrow$) | LPIPS ($\downarrow$) | DINO ($\uparrow$) |
> | - | - |:-:|:-:|:-:|
> |Block 4|rMLE|0.4888|0.4198|0.7776|
> ||LM   |0.5855|0.2576|0.9012|
> ||GLASS|0.4872|0.3568|0.7315|
> ||DRAG |**0.7896**|**0.0898**|**0.9622**|
> |Block 5|rMLE |0.3980|0.5006|0.6739|
> ||LM   |0.4432|0.3409|0.7614|
> ||GLASS|0.2917|0.4223|0.6811|
> ||DRAG |**0.5206**|**0.2231**|**0.9001**|
>
> ---
> > The experimental results lack discussion. The method is more effective at reconstructing data from deep layers compared to shallow layers. What accounts for this difference?
>
> Optimization-based DRAs optimize the sample $x$ by minimizing $d_\mathcal{H}$. When the split point is deep, this primarily guides $x$ to align with the high-level features of the target image, without necessarily ensuring a match of pixel-level accuracy. Prior DRAs often fail in reconstructing the image from deep-layer IR because they do not sufficiently restrict the search space. To improve DRA, we leverage a diffusion prior to constrain $x$, based on the assumption that the private image is an natural image.
>
> ---
> > The paper should include the discussion of FORA, etc,. methods.
>
> We agree that FORA and related works are important to discuss in the context of privacy threats in split inference, and we will include a discussion of these methods in the revised paper.
>
> These works consider privacy risk under a different configuration from ours by exploring query-free data reconstruction attacks in split learning. In that setup, the attacker cannot directly access $f_c$ but may capture or interfere with the training process to build a surrogate model $\tilde{f}_c \approx f_c$. Once this surrogate model is built, attackers can reconstruct private data using either optimization-based (e.g., DRAG) or learning-based methods. Xu et al. [1] note that combining these two research areas can lead to more powerful reconstruction attacks, as their developments are independent.
>
> On the other hand, our work focuses on the privacy risks associated with using foundation models as part of the model parameters in downstream tasks, implying that an attacker can feasibly access $f_c$ directly. Our findings highlight the need to develop privacy-preserving inference techniques, especially as new applications [2, 3] increasingly leverage foundation models.
>
> ---
> We appreciate your feedback and remain available to address any additional questions or concerns you may have.
>
> ### Reference
> [1] Xu, X., et al. A stealthy wrongdoer: Feature-oriented reconstruction attack against split learning. In CVPR, 2024.
>
> [2] Liu, H., et al. Visual instruction tuning. In NeurIPS, 2023.
>
> [3] Chen, J., et al. Minigpt-v2: large language model as a unified interface for vision-language multi-task learning. arXiv preprint arXiv:2310.09478. 2023
>
> [4] Wang, T., et al. Improving robustness to model inversion attacks via mutual information regularization. In AAAI, 2021
>
> [5] Zou, T., et al. Mutual information regularization for vertical federated learning. arXiv preprint arXiv:2301.01142. 2023
>
> [6] Duan, L., et al. Reimagining Mutual Information for Enhanced Defense against Data Leakage in Collaborative Inference. In NeurIPS. 2024

---

### Decision · Program_Chairs · 2025-05-01

**Decision:**

Accept (poster)

**Comment:**

This paper proposes DRAG, a new data reconstruction attack against split inference. The key idea is to use a pre-trained diffusion model as the image prior, which significantly reduces the dimensionality of the feature inversion optimization problem. The authors show that DRAG is able to invert deeper layer representations compared to prior work, and can remain effective against token dropping and shuffling defenses.

Reviewers generally agreed that DRAG is novel and effective. Some reviewers suggested improving the writing to improve the clarity of technical details. The main point of contention is whether or not the experiments are sufficient enough to demonstrate DRAG's effectiveness in different scenarios, including:
1. Different pre-trained diffusion model image prior.
2. Different image encoder architectures, including CNN and ViT.
3. Comparison with prior work suggested by Reviewers u9u9, DMzW and i33r.

The authors responded to these weaknesses directly in the rebuttal, addressing Weakness 1 by using an OOD diffusion model, and addressing Weakness 2 by evaluating on RN50 image encoders. The authors also explained the difference in attack setting compared to prior work. Although reviewers were not fully convinced by the rebuttal experiments, AC believes the paper has shown sufficient evidence that DRAG is a superior data reconstruction attack compared to prior work, and is one of the first that effectively utilizes the diffusion image prior to do so. As a result, the paper's merits outweigh its weaknesses, and should be of considerable value to the ICML community going forward.